# FEATURE DISCRIMINATION ANALYSIS FOR BINARY AND TERNARY QUANTIZATION

## ABSTRACT

In machine learning, quantization is widely used to simplify data representation and facilitate algorithm deployment on hardware. Considering the fundamental role of classification in machine learning, it is imperative to investigate the impact of quantization on classification. Current research primarily revolves around quantization errors, under the assumption that higher quantization errors generally lead to lower classification performance. However, this assumption lacks a solid theoretical foundation, and often contradicts empirical findings. For instance, some extremely low bit-width quantization methods, such as $\{0, 1\}$-binary quantization and $\{0, \pm 1\}$-ternary quantization, can achieve comparable or even superior classification accuracy compared to the original non-quantized data, despite exhibiting high quantization errors. To evaluate the classification performance more accurately, we propose to directly investigate the feature discrimination of quantized data, rather than analyze its quantization error. It is found that both binary and ternary quantization methods can surprisingly improve, rather than degrade, the feature discrimination of original data. This remarkable performance is validated through classification experiments on diverse data types, including images, speech and text.

## 1 INTRODUCTION

Quantization has been widely applied in machine learning to simplify data storage and computation complexities, while also catering to the requirements of algorithm deployment on digital hardware. In general, this operation will lead to a decrease in classification accuracy (Baras & Dey, 1999; Hoefler et al., 2021), due to reducing the precision of data or model parameters. To achieve a balance between complexity and accuracy, it is crucial to delve into the impact of quantization on classification. Currently, the impact is mainly evaluated through quantization errors, with the assumption that larger quantization errors generally lead to decreased classification accuracy (Lin et al., 2016a). However, this assumption lacks a solid theoretical basis (Lin et al., 2016a), as it merely adopts the quantization principle from signal processing (Gray & Neuhoff, 1998), which primarily focuses on data reconstruction fidelity rather than classification accuracy. In practice, it seems challenging to accurately assess the classification performance solely based on quantization errors.

For instance, it has been observed that some extremely low bit-width quantization methods, such as the $\{0, 1\}$-binary quantization and $\{0, \pm 1\}$-ternary quantization, which have been successively applied large-scale retrieval (Charikar, 2002) and deep network quantization (Qin et al., 2020; Gholami et al., 2022), can achieve comparable or even superior classification performance than their full-precision counterparts (Courbariaux et al., 2015; Lin et al., 2016b; Lu et al., 2023), despite suffering from high quantization errors. Apparently, the remarkable classification improvement resulting from quantization should not be attributed to the high quantization errors. This reveals the inadequacy of quantization errors in assessing the actual classification performance. Due to the absence of a theoretical explanation, the classification improvement induced by quantization has often been regarded as incidental and received little attention. Instead of quantization errors, in the paper we demonstrate that this intriguing phenomenon can be reasonably explained by feature discrimination. Following the Fisher's linear discriminant analysis (Fisher, 1936), we here refer to feature discrimination as the ratio between inter-class and intra-class scatters, and evaluate the classification

performance based on the rule that the higher the feature discrimination, the easier the classification. To the best of our knowledge, this is the first study that exploits feature discrimination to analyze the impact of quantization on classification, although it is more direct and reasonable than quantization errors in evaluating classification performance. The scarcity of relevant research can be attributed to the nonlinearity of the quantization operation, which substantially increases the analytical complexity of feature discrimination functions.

In the paper, it is demonstrated that the impact of the threshold-based binary and ternary quantization on feature discrimination can be analyzed, when the data are appropriately modeled using a Gaussian mixture model, with each Gaussian element representing one class of data. The Gaussian mixture model is chosen here based on two considerations. Firstly, the model has been well-established for approximating the distributions of real-world data (Torralba & Oliva, 2003; Weiss & Freeman, 2007) and their feature transformations (Wainwright & Simoncelli, 1999; Lam & Goodman, 2000). Secondly, the closure property of Gaussian distributions under linear operations can simplify the analysis of the feature discrimination function. By analyzing the discrimination across varying quantization thresholds, it is found that there exist certain quantization thresholds that can improve the discrimination of original data, thereby yielding improved classification performance. This finding is extensively validated through classification experiments both on synthetic and real data.

The related works are discussed as follows. As mentioned earlier, our work should be the first to take advantage of feature discrimination to investigate the impact of quantization on classification. In the filed of signal processing, there have been a few works proposed to reduce the negative impact of quantization on signal detection or classification (Poor & Thomas, 1977; Oehler & Gray, 1995). However, these studies did not employ feature discrimination analysis, distinguishing them from our research in both methodology and outcomes. Specifically, in these studies the model design accounts for both reconstruction loss and classification loss. The classification loss is primarily modeled in several ways, such as directly minimizing the classification error on quantized data (Srinivasamurthy & Ortega, 2002), enlarging the inter-class distance between quantized data (Jana & Moulin, 2000; 2003), reducing the difference between the distributions of quantized data and original data (Baras & Dey, 1999), as well as minimizing the discrepancy in classification before and after quantization (Dogahe & Murthi, 2011). Through analyses of these losses, the classification performance of quantized data can only approach, rather than surpass, the performance of original data (Baras & Dey, 1999).

## 2 PROBLEM FORMULATION

In this section, we specify the feature discrimination functions for the original (non-quantized) and quantized data. Prior to this, we introduce the binary and ternary quantization functions, as well as the data modeling.

### 2.1 QUANTIZATION FUNCTIONS

The binary and ternary quantization functions are formulated as

$$f_b(x; \tau) = \begin{cases} 1, & \text{if } x > \tau \\ 0, & \text{otherwise} \end{cases} \tag{1}$$

and

$$f_t(x; \tau) = \begin{cases} 1, & \text{if } x > \tau \\ 0, & \text{if } -\tau \le x \le \tau \\ -1, & \text{if } x < -\tau \end{cases} \tag{2}$$

where the threshold parameter $\tau \in (-\infty, +\infty)$ for $f_b(x; \tau)$, and $\tau \in [0, +\infty)$ for $f_t(x; \tau)$. The two functions operate element-wise on a vector $\mathbf{x} = [x_1, x_2, \cdots, x_n]^\top \in \mathbb{R}^n$, namely $f_b(\mathbf{x}; \tau) = (f_b(x_1; \tau), f_b(x_2; \tau), \cdots, f_b(x_n; \tau))^\top$ and the same applies to $f_t(\mathbf{x}; \tau)$.

### 2.2 DATA DISTRIBUTIONS

Throughout the work, we denote each data sample using a vector. For the sake of generality, as discussed before, we assume that the data vector randomly drawn from a class is a random vector

$\mathbf{X} = \{X_1, X_2, \cdots, X_n\}^\top$, with its each element $X_i$ following a Gaussian distribution $N(\mu_{1,i}, \sigma^2)$; and similarly, for the random vector $\mathbf{Y} = \{Y_1, Y_2, \cdots, Y_n\}^\top$ drawn from another class, we suppose its each element $Y_i \sim N(\mu_{2,i}, \sigma^2)$, where $\mu_{2,i} \neq \mu_{1,i}$. Considering that the discrimination between the two random vectors $\mathbf{X}$ and $\mathbf{Y}$ positively correlates with the discrimination between their each pair of corresponding elements $X_i$ and $Y_i$, we propose to analyze the discrimination at the element level, specifically between $X_i$ and $Y_i$, rather than between the entire vectors, $\mathbf{X}$ and $\mathbf{Y}$. For notational convenience, without causing confusion, in the sequel we will omit the subscript "$i$" of $X_i$ and $Y_i$, and write their distributions as $X \sim N(\mu_1, \sigma^2)$ and $Y \sim N(\mu_2, \sigma^2)$, where $\mu_1 \neq \mu_2$. Note that we assume here that the two variables $X$ and $Y$ share the same variance $\sigma^2$. This assumption is common in statistical research, as the data we intend to investigate are often drawn from the same or similar scenarios and thus exhibit similar noise levels.

When standardization, a common practice in machine learning, is applied to the two variables, $X$ and $Y$, their distributions will exhibit specific relationships. More precisely, in a binary classification problem, the dataset we handle is a mixture, denoted as $Z$, comprising two classes of samples drawn respectively from $X$ and $Y$. Usually, the mixture $Z$ is assumed to possess a balanced class distribution, meaning that samples are drawn from $X$ and $Y$ with equal probabilities. Under this assumption, when we perform standardization by subtracting the mean and dividing by the standard deviation for each sample in $Z$, the distributions of $X$ and $Y$ (in $Z$) will become

$$\tilde{X} = \frac{X - E[Z]}{\sqrt{D[Z]}} \sim N\left(\frac{(\mu_1 - \mu_2)/2}{\sqrt{\sigma^2 + \frac{1}{4}(\mu_1 - \mu_2)^2}}, \frac{\sigma^2}{\sigma^2 + \frac{1}{4}(\mu_1 - \mu_2)^2}\right) \tag{3}$$

and

$$\tilde{Y} = \frac{Y - E[Z]}{\sqrt{D[Z]}} \sim N\left(\frac{-(\mu_1 - \mu_2)/2}{\sqrt{\sigma^2 + \frac{1}{4}(\mu_1 - \mu_2)^2}}, \frac{\sigma^2}{\sigma^2 + \frac{1}{4}(\mu_1 - \mu_2)^2}\right) \tag{4}$$

where $E[Z]$ and $D[Z]$ denote the expectation and variance of $Z$, which have expressions $E[Z] = \frac{1}{2}(\mu_1 + \mu_2)$ and $D[Z] = \sigma^2 + \frac{1}{4}(\mu_1 - \mu_2)^2$.

From Equations (3) and (4), it can be seen that after standardization, the two classes of variables $\tilde{X}$ and $\tilde{Y}$ still exhibit Gaussian distributions, but showcase two interesting properties: 1) their means are symmetric about zero; and 2) they have the sum of the square of the mean and the variance equal to one. By the two properties, the distributions of two classes of standardized data are characterized in Property 1. In the paper, we will typically focus our study on the standardized data.

**Property 1** (The distributions of two classes of standardized data). The two classes of standardized data we aim to study have their samples i.i.d drawn from $X \sim N(\mu, \sigma^2)$ and $Y \sim N(-\mu, \sigma^2)$, where $\mu^2 + \sigma^2 = 1$, $\mu \in (0, 1)$.

## 2.3 FEATURE DISCRIMINATION

Following the Fisher's linear discriminant rule, we define the discrimination between two classes of data as the ratio of the expected inter-class distance to the expected intra-class distance, as specified below.

**Definition 1** (Discrimination between two classes of data). For two classes of data with samples respectively drawn from the variables $X$ and $Y$, the discrimination between them is defined as

$$D = \frac{E[(X_1 - Y_1)^2]}{E[(X_1 - X_2)^2] + E[(Y_1 - Y_2)^2]} \tag{5}$$

where $X_1$ and $X_2$ are i.i.d. samples of $X$, and $Y_1$ and $Y_2$ are i.i.d samples of $Y$.

In the sequel, we will utilize the above definition $D$ to denote the discrimination between original (non-quantized) data; and for the binary and ternary quantized data, as detailed below, we adopt $D_b$ and $D_t$ to represent their discrimination.

**Definition 2** (Discrimination between two classes of quantized data). Following the discrimination specified in Definition 1, the discrimination between two binary quantized data $X_b = f_b(X; \tau)$ and

$Y_b = f_b(Y; \tau)$, is formulated as

$$D_b = \frac{E[(X_{1,b} - Y_{1,b})^2]}{E[(X_{1,b} - X_{2,b})^2] + E[(Y_{1,b} - Y_{2,b})^2]} \tag{6}$$

where $X_{1,b}$ and $X_{2,b}$ are i.i.d. samples of $X_b$, and $Y_{1,b}$ and $Y_{2,b}$ are i.i.d. samples of $Y_b$. Similarly, the discrimination between two ternary quantized data $X_t = f_t(X; \tau)$ and $Y_t = f_t(Y; \tau)$ is expressed as

$$D_t = \frac{E[(X_{1,t} - Y_{1,t})^2]}{E[(X_{1,t} - X_{2,t})^2] + E[(Y_{1,t} - Y_{2,t})^2]} \tag{7}$$

where $X_{1,t}$ and $X_{2,t}$ are i.i.d. samples of $X_t$, and $Y_{1,t}$ and $Y_{2,t}$ are i.i.d. samples of $Y_t$.

## 2.4 GOAL

The major goal of the paper is to investigate whether there exist threshold values $\tau$ in the binary quantization $f_b(x; \tau)$ and the ternary quantization $f_t(x; \tau)$, such that the quantization can improve the feature discrimination of original data, namely having $D_b > D$ and $D_t > D$.

## 3 DISCRIMINATION ANALYSIS

### 3.1 THEORETICAL RESULTS

**Theorem 1** (Binary Quantization). Consider the discrimination $D$ between two classes of data $X \sim N(\mu, \sigma^2)$ and $Y \sim N(-\mu, \sigma^2)$ as specified in Property 1, as well as the discrimination $D_b$ between their binary quantization $X_b = f_b(X; \tau)$ and $Y_b = f_b(Y; \tau)$. We have $D_b > D$, if there exists a quantization threshold $\tau \in (-\infty, +\infty)$ such that

$$\beta - \alpha + \frac{\mu^2(1 - 2\beta) - \mu\sqrt{\mu^2 + 4\beta(1 - \beta)}}{1 + \mu^2} > 0, \tag{8}$$

where $\alpha = \Phi\left(\frac{\tau - \mu}{\sigma}\right)$ and $\beta = \Phi\left(\frac{\tau + \mu}{\sigma}\right)$, with $\Phi(\cdot)$ denoting the cumulative distribution function of the standard normal distribution.

**Theorem 2** (Ternary Quantization). Consider the discrimination $D$ between two classes of data $X \sim N(\mu, \sigma^2)$ and $Y \sim N(-\mu, \sigma^2)$ as specified in Property 1, as well as the discrimination $D_t$ between their ternary quantization $X_t = f_t(X; \tau)$ and $Y_t = f_t(Y; \tau)$. We have $D_t > D$, if there exists a quantization threshold $\tau \in [0, +\infty)$ such that

$$\beta - \alpha + \frac{\mu^2 - \sqrt{\mu^4 + 8\mu^2\beta}}{2} > 0, \tag{9}$$

where $\alpha = \Phi\left(\frac{-\tau - \mu}{\sigma}\right)$ and $\beta = \Phi\left(\frac{-\tau + \mu}{\sigma}\right)$, with $\Phi(\cdot)$ denoting the cumulative distribution function of the standard normal distribution.

**Remarks:** Regarding the two theorems, there are three issues worth discussing. 1) The two theorems suggest that both binary and ternary quantization methods can indeed improve the classification performance of original data, if there exist quantization thresholds $\tau$ that can satisfy the constraints shown in Equations (8) and (9). The following numerical analysis demonstrates that the desired threshold $\tau$ does exist, when the two classes of data $X \sim N(\mu, \sigma^2)$ and $Y \sim N(-\mu, \sigma^2)$ are assigned appropriate values for $\mu$ and $\sigma$. This threshold $\tau$ can be approximately estimated using the bisection method. 2) Our theoretical analysis is based on the premise that the data vectors belonging to the same class have Gaussian distributions at the vectors' each coordinate. This condition should hold true when two classes of data are readily separable, as in this case the data points within each class should cluster tightly, allowing for Gaussian approximation. This explains the recent research findings, that the binary or ternary quantization tends to achieve comparable or superior classification performance, when handling relatively simple datasets (Courbariaux et al., 2015; Lin et al., 2016b), or distinguishable features (Lu et al., 2023). 3) The conclusion we derive in Theorem 1 for $\{0, 1\}$-binary quantization also applies to another popular $\{-1, 1\}$-binary quantization (Qin et al., 2020), since the Euclidean distance of the former is equivalent to the cosine distance of the latter.

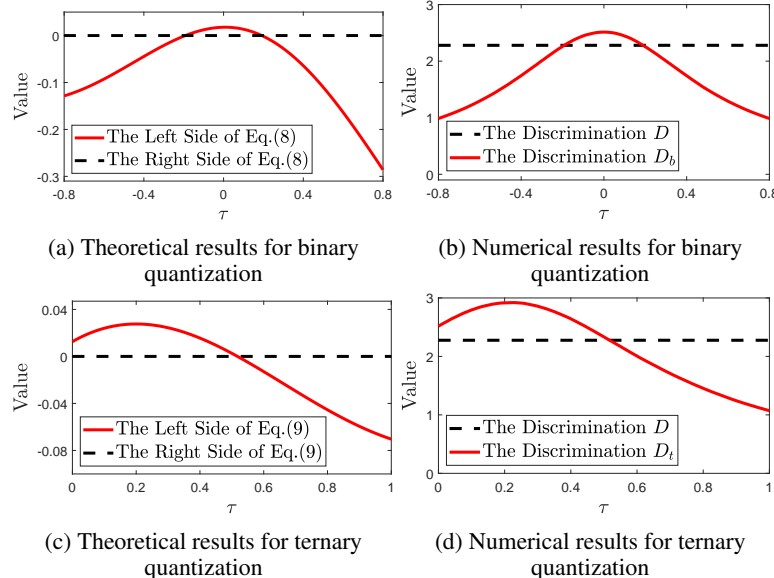

Figure 1: Consider two classes of data $X \sim N(\mu, \sigma^2)$ and $Y \sim N(-\mu, \sigma^2)$, with $\mu = 0.8$ and $\sigma^2 = 0.36$, as specified in Property 1. The values for the left and right sides of Equations (8) and (9) are provided in (a) and (c) for binary and ternary quantization, respectively; and the discrimination $D$, $D_b$ and $D_t$ statistically estimated with Equations (5), (6) and (7) are illustrated in (b) and (d) for binary and ternary quantization, respectively.

## 3.2 NUMERICAL ANALYSIS

In this part, we conduct numerical analyses for two primary objectives. Firstly, we aim to prove the existence of the desired quantization threshold $\tau$ that holds Equations (8) and (9), namely making the left sides of the two inequalities larger than their right sides (with values equal to zero). For this purpose, we compute the values of the left sides of Equations (8) and (9), through assigning specific values to $\tau$, as well as to the two variables $X$ and $Y$'s distribution parameters $\mu$ and $\sigma^2$. Note that we here set $\sigma^2 = 1 - \mu^2$, $\mu \in (0, 1)$, in accordance with Property 1. In Figure 1, we examine the case that fixes $\mu = 0.8$ and $\sigma^2 = 0.36$, while varying the value of $\tau$ with a step width 0.01. The results for binary quantization and ternary quantization are provided in Figures 1 (a) and (c), respectively. It can be seen that for the two quantization methods, their conditions shown in Equations (8) and (9) will hold when respectively having $\tau \in [-0.2, 0.2]$ and $\tau \in [0, 0.5]$. This proves the existence of the desired quantization threshold $\tau$ that can improve feature discrimination. For limited space, we here only discuss the case of $\mu = 0.8$ (and $\sigma^2 = 1 - \mu^2$) in Figure 1. By examining different $\mu \in (0, 1)$ in the same way, we can find that the quantization threshold $\tau$ that holds Equations (8) and (9), is present when $\mu \in (0.76, 1)$ and $\mu \in (0.66, 1)$, respectively; see Figures 7 and 8 for more evidences. This result implies two consequences. On one hand, ternary quantization has more chances to improve feature discrimination compared to binary quantization, as the former has a broader range of $\mu$. On the other hand, the improved discrimination tends to be achieved when $\mu$ is sufficiently large, coupled with a correspondingly small $\sigma$, or when the discrimination between two classes of data is sufficiently high. Empirically, as depicted in Figure 17, the two specific ranges of $\mu$ values are attainable for the commonly-used features of real data.

The second goal is to verify that the quantization thresholds $\tau$ we estimate with Equations (8) and (9) in Theorems 1 and 2, can indeed improve feature discrimination. To this end, it needs to prove that the ranges of $\tau$ derived by Equations (8) and (9), such as the ones depicted in Figures 1 (a) and (c), are consistent with the ranges we can statistically estimate by the discrimination definitions $D$, $D_b$ and $D_t$, as specified in Definitions 1 and 2. To estimate the discrimination $D$, $D_b$ and $D_t$, we randomly generate 10,000 samples from $X \sim N(0.8, 0.36)$ and $Y \sim N(-0.8, 0.36)$, respectively, and then statistically estimate them with Equations (5), (6) and (7), across varying $\tau$ (with a step width 0.01). The results are provided in Figures 1 (b) and (d), respectively for binary quantization and ternary quantization. It can be seen that we have $\tau \in [-0.2, 0.2]$ for $D_b > D$, and have

$\tau \in [0, 0.5]$ for $D_t > D$. The results coincide with the theoretical results shown in Figures 1 (a) and (c), validating the correctness of Theorems 1 and 2.

## 4 EXPERIMENTS

Through previous theoretical and numerical analyses, we have demonstrated that binary and ternary quantization can improve feature discrimination between two classes of data, when the data vectors within each class exhibit Gaussian distributions across each coordinate point of their feature vectors. Given that improved feature discrimination should yield better classification performance, this section aims to validate this improvement by assessing classification results.

Our experiments will mainly investigate binary classification using two fundamental linear classifiers: the $k$-nearest neighbors (KNN) algorithm (with $k = 5$) (Peterson, 2009), employing both Euclidean and cosine distances as similarity metrics, and the support vector machine (SVM) (Cortes & Vapnik, 1995), equipped with a linear kernel. We choose linear classifiers for binary classification based on two considerations. Firstly, linear binary classification can directly reflect the feature discrimination between two classes, unlike more complex nonlinear classifiers that often involve feature selection operations. Secondly, linear binary classification is a foundational concept in machine learning. The insights gained from this analysis can be extended to multiclass and nonlinear classifier-based classifications, as evidenced in the subsequent experiments.

To assess the robustness and generalizability of our theoretical findings, we will conduct classification evaluations on both synthetic and real data. Synthetic data can conform perfectly to the distribution conditions outlined in our theoretical analysis, whereas real data usually cannot.

### 4.1 SYNTHETIC DATA

#### 4.1.1 SETTING

In the simulation, we suppose that two classes of data have their samples i.i.d. drawn from two different random vectors $\mathbf{X} = \{X_1, X_2, \cdots, X_n\}^\top$ and $\mathbf{Y} = \{Y_1, Y_2, \cdots, Y_n\}^\top$, for which we set $X_i \sim N(\mu_i, \sigma_i^2)$ and $Y_i \sim N(-\mu_i, \sigma_i^2)$, with $\mu_i \in (-1, 0) \cup (0, 1)$ and $\sigma_i^2 = 1 - \mu_i^2$, in accordance with the data distributions specified in Property 1. Considering the fact that the features of real-world data usually exhibit sparse structures (Weiss & Freeman, 2007; Kotz et al., 2012), we further suppose that the means $\mu_i$ decay exponentially in magnitude, i.e. $|\mu_{i+1}|/|\mu_i| = e^{-\lambda}$, $\lambda \geq 0$, and set $\mu_1 = 0.8$ in the following simulation. It can be seen that with the increasing of $\lambda$, the mean's magnitude $|\mu_i|$ (with $i > 1$) will become smaller, indicating a smaller data element $X_i$ (in magnitude) and a sparser data structure. However, the data element $X_i$ with smaller $\mu_i$, is not favorable for quantization to improve feature discrimination, as indicated by previous numerical analyses. The impact of data sparsity on quantization can be investigated by increasing the value of the parameter $\lambda$.

With the data model described above, we randomly generate two classes of data, each class containing 1000 samples. The dataset is split into two parts for training and testing, in a ratio of 4:1. Then we evaluate the KNN and SVM classification on them. The classification accuracy is determined by averaging the accuracy results obtained from repeating the data generation and classification process 100 times. The results for KNN with Euclidean distance are provided in Figures 2 and 3, and the results for KNN with cosine distance and SVM are given in the appendix, Figures 9–12. It can be seen that the three classifiers exhibit similar performance trends. For conciseness, we will focus more on the results of KNN with Euclidean distance in the following discussion.

#### 4.1.2 RESULTS

**Comparison between the data with different sparsity.** In Figure 2, we investigate the classification performance for the data generated with different parameters $\lambda \in \{0, 0.01, 0.1, 1\}$, namely with different sparsity levels. Recall that the larger the $\lambda$, the smaller the $|\mu_i|$, or say the smaller the data element $X_i$ (in magnitude). By previous analyses, the data element $X_i$ with smaller $|\mu_i|$ is not conducive to enhancing feature discrimination through quantization. Nevertheless, empirically, the negative effect does not appear to be significant. From Figure 2, it can be seen that when increasing

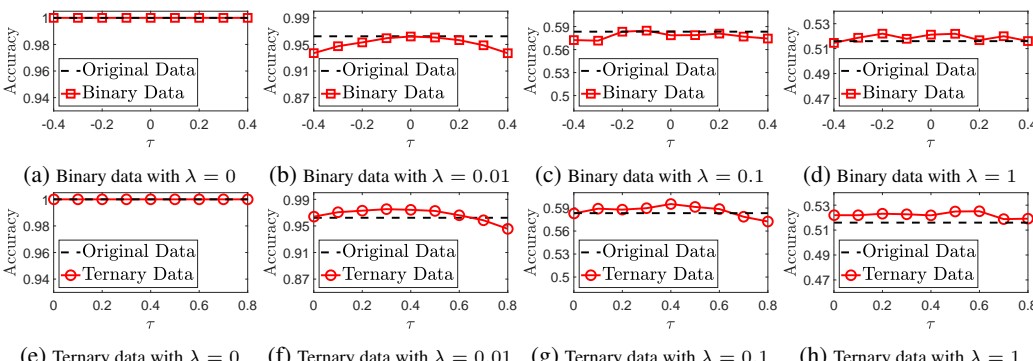

**Figure 2:** KNN (Euclidean distance) classification accuracy for the 10,000-dimensional binary, ternary, and original data that are generated with the varying parameter $\lambda \in \{0, 0.01, 0.1, 1\}$, which controls the data sparsity.

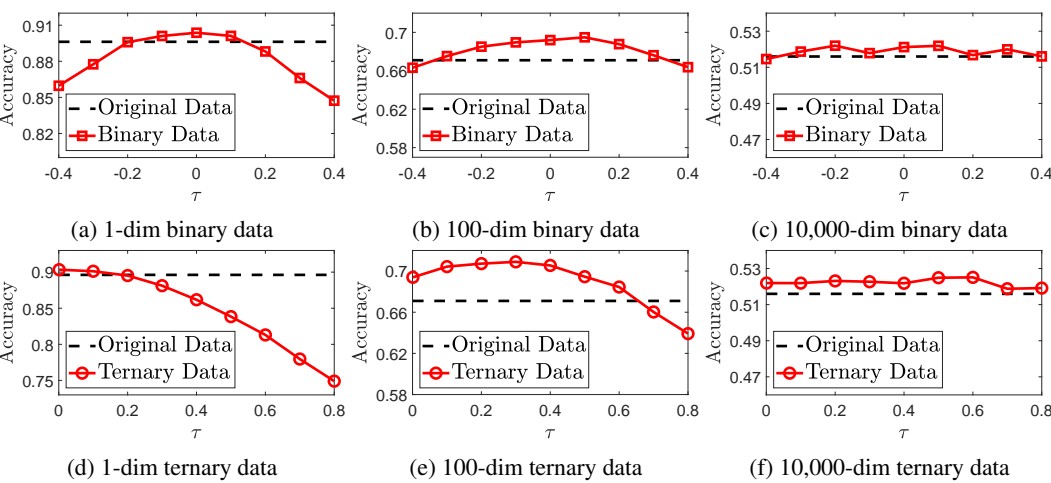

**Figure 3:** KNN (Euclidean distance) classification accuracy for the binary, ternary, and original data generated with the parameter $\lambda = 1$, and with varying dimensions $n \in \{1, 100, 10000\}$.

$\lambda$ from 0.1 to 1, there have been quantization thresholds $\tau$ that can yield better classification performance than original data. In addition, it noteworthy that as $\lambda$ increases, the overall classification accuracy of original data will decrease. This decreasing trend also impacts the absolute performance of the quantized data, even though it may outperform original data.

**Comparison between the data with different dimensions.** The impact of data dimensions $n \in \{1, 100, 10000\}$ on classification is investigated in Figure 3, where the data are generated with the exponentially decaying parameter $\lambda = 1$. It can be seen that with the increasing of data dimension, the range of the quantization thresholds $\tau$ that outperform original data tends to expand, but the performance advantage declines. As previously discussed, the decline should be attributed to the data element $X_i$ with small means $|\mu_i|$, whose quantity will rise with the data dimension $n$, particularly when the decay parameter $\lambda$ of $|\mu_i|$ is large. To alleviate this adverse effect, it is recommended to choose a relatively smaller $\lambda$ for high-dimensional data, indicating a structure that is not overly sparse. Conversely, when the high-dimensional data is highly sparse, we should reduce its dimension to improve the classification performance under quantization.

**Comparison between binary quantization and ternary quantization.** From Figures 2 and 3, it can be seen that ternary quantization surpasses binary quantization by offering broader ranges of quantization thresholds $\tau$ that can yield higher classification accuracy than original data. This observation is consistent with our previous theoretical and numerical analyses.

**Comparison between KNN and SVM.** Combining the results in Figures 2, 3, and 9–12, we can say that both KNN and SVM enable quantization to improve the classification accuracy of original data, within specific ranges of quantization thresholds $\tau$. If closely examining these ranges, it can be observed that KNN often performs better when using Euclidean distance than using cosine distance. This can be attributed to the advantage of Euclidean distance over cosine distance in measuring the distance between 0 and $\pm 1$. Also, KNN often outperforms SVM, such as the case of $\lambda = 0.1$ shown in Figures 2 and 10. This is because the support vector of SVM relies on a few data points located on the boundary between two classes, which may deteriorate during quantization. In contrast, KNN depends on the high-quality data points within each class, making it resilient to quantization noise.

**Comparison between classification accuracy, feature discrimination and quantization error.** Figure 16 illustrates that the classification accuracy of quantized data across varying quantization threshold $\tau$ can be reasonably reflected by feature discrimination, rather than quantization errors.

### 4.2 REAL DATA

#### 4.2.1 SETTING

The classification is conducted on three different types of datasets, including the image datasets YaleB (Lee et al., 2005), CIFAR10 (Krizhevsky & Hinton, 2009) and ImageNet1000 (Deng et al., 2009), the speech dataset TIMIT (Fisher et al., 1986), and the text dataset Newsgroup (Lang, 1995). The datasets are briefly introduced as follows. YaleB contains face images of 38 persons, with about 64 faces per person. CIFAR10 consists of 60,000 color images from 10 different classes, with each class having 6,000 images. ImageNet1000 consists of 1000 object classes, with 1,281,167 training images, 50,000 validation images, and 100,000 test images. For the above three image datasets, we separately extract their features using Discrete Wavelet Transform (DWT), ResNet18 (He et al., 2016) and VGG16 (Simonyan & Zisserman, 2014). For ease of simulation, the resulting feature vectors are dimensionally reduced by integer multiples, leading to the sizes of 1200, 5018, and 5018 respectively. From TIMIT, as in (Mohamed et al., 2011; Hutchinson et al., 2012), we extract 39 classes of 429-dimensional phoneme features for classification, totally with 1,134,138 training samples and 58,399 test samples. Newsgroup comprises 20 categories of texts, with 11,269 samples for training, and 7,505 samples for testing. The feature dimension is reduced to 5000 by selecting the top 5000 most frequent words in the bag of words, as done in (Larochelle et al., 2012).

For each dataset, we iterate through all possible class pairs to perform binary classification. The samples for training and testing are selected according to the default settings of the datasets. For YaleB without prior settings, we randomly assign half of the samples for training and the remaining half for testing. In the simulation, we need to test the classification performance of quantized data across varying quantization threshold $\tau$. The value of $\tau$ should correlate with the element scale of the feature vectors, in the pursuit of improving classification over orginal data. To address the scale varying of $\tau$ across different data, we here suppose that $\tau = \gamma \cdot \eta$, where $\eta$ denotes the average magnitude of the feature elements (coordinates) in all the feature vectors used for classification, and $\gamma$ is a scaling parameter. By adjusting $\gamma$ within a narrow range, as illustrated later, we can derive the desired $\tau$ for various types of data.

To verify the generalizability of our feature discrimination analysis between two classes, we not only evaluate binary classification using KNN and SVM, but also conduct multiclass classification, as well as nonlinear classification using multilayer perceptron (MLP) (Rumelhart et al., 1986) and decision trees (Quinlan, 1986). Due to space limitations, in the main body, we present the classification results of YaleB, Newsgroup, and TIMIT using KNN with Euclidean distance and SVM, as illustrated in Figures 4 to 6. The results for other datasets, such as CIFAR10 and ImageNet1000, and other classifiers, including KNN with cosine distance, MLP and decision trees, are provided in the appendix, but briefly discussed within the main text.

#### 4.2.2 RESULTS

**Binary classification using KNN and SVM.** From Figures 4-6,13, 14 and 21, it can be seen that that within specific ranges of quantization thresholds $\tau$, both binary and ternary quantization

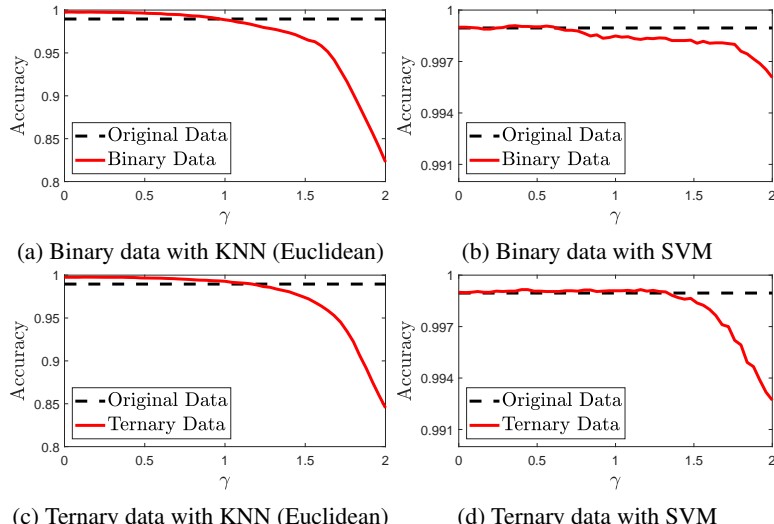

Figure 4: Classification accuracy for the binary, ternary, and original data by KNN (Euclidean distance) and SVM on YaleB. The parameter $\gamma$ corresponds to a threshold $\tau = \gamma \cdot \eta$, where $\eta$ denotes the average magnitude of the feature elements in all feature vectors.

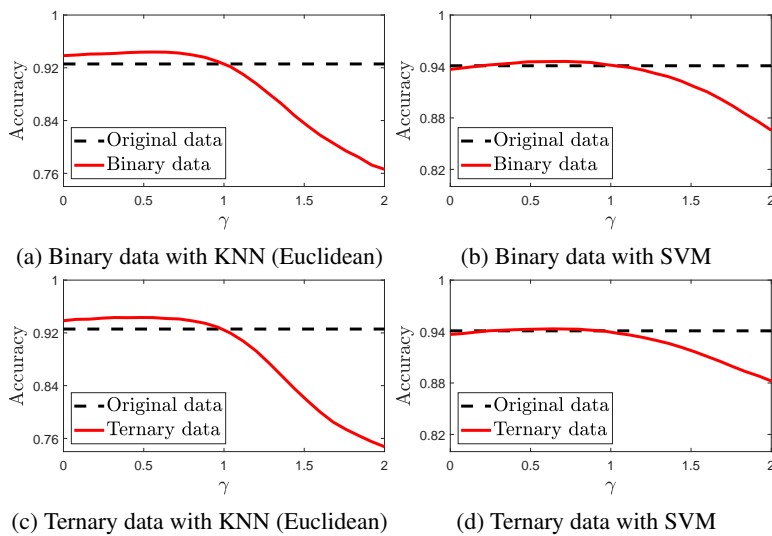

Figure 5: Classification accuracy for the binary, ternary, and original data by KNN (Euclidean distance) and SVM on TIMIT. The parameter $\gamma$ corresponds to a threshold $\tau = \gamma \cdot \eta$, where $\eta$ denotes the average magnitude of the feature elements in all feature vectors.

can achieve superior or at least equivalent classification performance compared to the original data across five different datasets, although as shown in Figure 17, each data class does not adequately conform to the Gaussian distribution assumption underlying our theoretical analysis. Similarly as in the classification of synthetic data, we observe the following results. Firstly, when using Euclidean distance, KNN consistently identifies quantization thresholds $\tau$ that improve the classification of original data across all datasets. Secondly, compared to cosine distance, Euclidean distance tends to allow KNN to encompass a wider range of $\tau$ values that are beneficial for improving classification. Thirdly, with SVM, quantization occasionally achieves comparable performance to the original data, rather than surpassing it, as exemplified in Figure 13. Fourthly, ternary quantization often outperforms binary quantization by providing a broader range of threshold $\tau$ values that facilitate classification improvement. The rationale behind these results has been elaborated in our

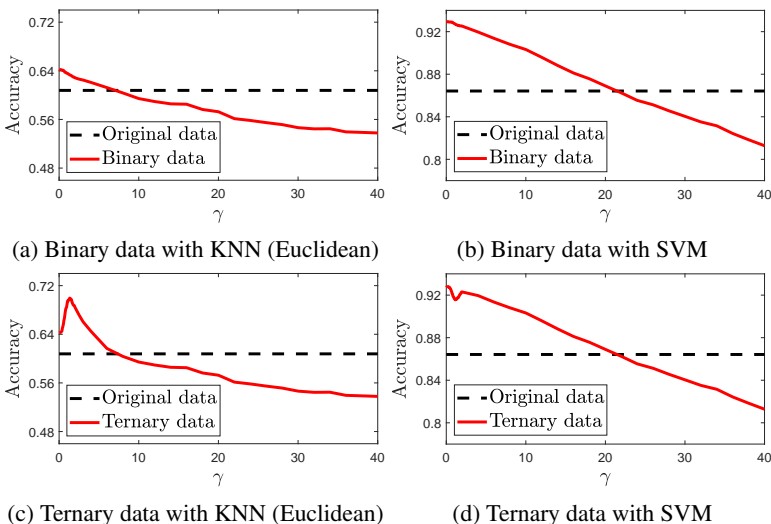

Figure 6: Classification accuracy for the binary, ternary, and original data by KNN (Euclidean distance) and SVM on Newsgroup. The parameter $\gamma$ corresponds to a threshold $\tau = \gamma \cdot \eta$, where $\eta$ denotes the average magnitude of the feature elements in all feature vectors.

previous classification analysis of synthetic data. The consistent performance observed in both real and synthetic data underscores the broad applicability of our theoretical findings.

**Multiclass and nonlinear classification.** While our feature discrimination analysis is focused on linear, binary classification, experiments demonstrate that our results can also be extended to multiclass and nonlinear classifications. For example, in multiclass classification on ImageNet1000, quantization thresholds $\tau$ that improve the classification of original data have been identified, as shown in Figure 22. The extension from binary to multiclass classification may be explained by the fact that feature elements sharing a common coordinate (or feature attribute) across different classes tend to exhibit a binary state: strong or weak, as illustrated in Figure 18, which indicates the intensity of the feature attribute within a feature vector. This suggests that multiclass classification at each feature coordinate can be viewed as a binary classification problem. Figures 19 and 20 demonstrate that the desired thresholds $\tau$ can also be obtained in nonlinear classifications using MLP and decision trees. The extension from linear to nonlinear classification may be attributed to the fundamental linear operations often involved in nonlinear classifiers, which assess the linear discrimination between features or model parameters.

## 5 CONCLUSION

In the paper, we have proposed utilizing feature discrimination to analyze the impact of quantization on classification. Unlike traditional analyses, which are primarily based on quantization errors, our feature discrimination-based approach offers a more direct and reasoned assessment of classification performance. Through our analysis, we demonstrate that common binary and ternary quantization methods can improve the feature discrimination of original data, particularly when data vectors within the same class follow Gaussian distributions at each coordinate. This improved discrimination is validated through binary classification experiments on both synthetic and real data. While our feature discrimination analysis primarily focuses on linear, binary classification issues, our experiments indicate that the findings can be extended to multiclass and nonlinear classification scenarios. This underscores the broad applicability of our theoretical results. Importantly, our study challenges the traditional belief that larger quantization errors generally lead to lower classification performance, laying a theoretical foundation for developing better quantization methods.

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

## A  DETAILED PROOF

### A.1  PROOF OF THEOREM 1

Let $X_1$ and $X_2$ be i.i.d. samples of $X$, and $Y_1$ and $Y_2$ be i.i.d. samples of $Y$. Denote $X_{i,b}$ and $Y_{i,b}$ as the binary quantization of $X_i$ and $Y_i$, i.e. $X_{i,b} = f_b(X_i; \tau)$ and $Y_{i,b} = f_b(Y_i; \tau)$, where $i = 1, 2$. By the distributions of $X$ and $Y$ specified in Property 1 and the binary quantization function $f_b(x; \tau)$ defined in Equation(1), the probability mass functions of $X_{i,b}$ and $Y_{i,b}$ can be derived as

$$P(X_{i,b} = k) = \begin{cases} 1 - \alpha, & k = 1 \\ \alpha, & k = 0 \end{cases} \tag{10}$$

and

$$P(Y_{i,b} = k) = \begin{cases} 1 - \beta, & k = 1 \\ \beta, & k = 0 \end{cases} \tag{11}$$

where $\alpha = \Phi\left(\frac{\tau-\mu}{\sigma}\right)$ and $\beta = \Phi\left(\frac{\tau+\mu}{\sigma}\right)$. By the probability functions, it is easy to deduce that

$$
\begin{aligned}
E\left[(X_1 - X_2)^2\right] &= 2\sigma^2, & E\left[(X_{1,b} - X_{2,b})^2\right] &= 2\alpha - 2\alpha^2, \\
E\left[(Y_1 - Y_2)^2\right] &= 2\sigma^2, & E\left[(Y_{1,b} - Y_{2,b})^2\right] &= 2\beta - 2\beta^2, \\
E\left[(X_1 - Y_2)^2\right] &= 2\sigma^2 + 4\mu^2, & E\left[(X_{1,b} - Y_{1,b})^2\right] &= \alpha + \beta - 2\alpha\beta.
\end{aligned}
$$

With these equations, the discrimination $D$ of original data, as specified in Definition 1, can be further expressed as

$$D = \frac{E[(X_1 - Y_1)^2]}{E[(X_1 - X_2)^2] + E[(Y_1 - Y_2)^2]} = \frac{\sigma^2 + 2\mu^2}{2\sigma^2}, \tag{12}$$

and similarly, the discrimination $D_b$ of binary quantized data, as specified in Definition 2, can be written as

$$D_b = \frac{E[(X_{1,b} - Y_{1,b})^2]}{E[(X_{1,b} - X_{2,b})^2] + E[(Y_{1,b} - Y_{2,b})^2]} = \frac{\alpha - 2\alpha\beta + \beta}{(2\alpha - 2\alpha^2) + (2\beta - 2\beta^2)}. \tag{13}$$

Next, we are ready to prove that $D_b > D$ under the condition (8). By Equations (12) and (13), it is easy to see that $D_b > D$ is equivalent to

$$(\sigma^2 + 2\mu^2)\alpha^2 - 2(\sigma^2\beta + \mu^2)\alpha + (\sigma^2 + 2\mu^2)\beta^2 - 2\mu^2\beta > 0. \tag{14}$$

This inequality can be viewed as a quadratic inequality in $\alpha$, which has the discriminant:

$$\Delta = 4\mu^4 + 16(1 - \beta)\mu^2\beta > 0.$$

By the above inequality, the inequality (14) holds when $\alpha \in (-\infty, \alpha_1) \cup (\alpha_2, +\infty)$, where

$$\alpha_1 = \beta + \frac{\mu^2(1 - 2\beta) - \mu\sqrt{\mu^2 + 4\beta(1 - \beta)}}{1 + \mu^2},$$

and

$$\alpha_2 = \beta + \frac{\mu^2(1 - 2\beta) + \mu\sqrt{\mu^2 + 4\beta(1 - \beta)}}{1 + \mu^2}. \tag{15}$$

Given (15), we can further derive $\alpha_2 > \beta$, since $\mu^2(1 - 2\beta) + \mu\sqrt{\mu^2 + 4\beta(1 - \beta)} > 0$. However, this result contradicts the conclusion that $\alpha < \beta$ we can derive with the probability mass functions shown in (10) and (11), mainly by the increasing property of $\Phi(\cdot)$. So the solution to the inequality (14) should be $\alpha \in (-\infty, \alpha_1)$, implying $\beta - \alpha + \frac{\mu^2(1-2\beta) - \mu\sqrt{\mu^2 + 4\beta(1-\beta)}}{1+\mu^2} > 0$.

### A.2  PROOF OF THEOREM 2

Let $X_1$ and $X_2$ be i.i.d. samples of $X$, and $Y_1$ and $Y_2$ be i.i.d. samples of $Y$. Denote $X_{i,t} = f_t(X_i; \tau)$ and $Y_{i,t} = f_t(Y_i; \tau)$, where $i = 1, 2$. By the distributions of $X$ and $Y$ specified in Property 1 and

the ternary quantization $f_t(x; \tau)$ defined in Equation (2), the probability mass functions of $X_{i,t}$ and $Y_{i,t}$ can be derived as

$$P(X_{i,t} = k) = \begin{cases} \beta, & k = 1 \\ 1 - \alpha - \beta, & k = 0 \\ \alpha, & k = -1 \end{cases} \tag{16}$$

$$P(Y_{i,t} = k) = \begin{cases} \alpha, & k = 1 \\ 1 - \alpha - \beta, & k = 0 \\ \beta, & k = -1 \end{cases} \tag{17}$$

where $\alpha = \Phi(\frac{-\tau - \mu}{\sigma})$ and $\beta = \Phi(\frac{-\tau + \mu}{\sigma})$.

Then, by Definition 2, the discrimination $D_t$ of ternary quantization can be derived as

$$D_t = \frac{E[(X_{1,t} - Y_{1,t})^2]}{E[(X_{1,t} - X_{2,t})^2] + E[(Y_{1,t} - Y_{2,t})^2]} = \frac{(\alpha + \alpha^2 - 2a\beta + \beta + \beta^2)}{2(\alpha - \alpha^2 + 2\alpha\beta + \beta - \beta^2)}. \tag{18}$$

By Equations (12) and (18), it can be seen that $D_t > D$ is equivalent to

$$\frac{(\alpha + \beta) + (\alpha - \beta)^2}{2(\alpha + \beta) - 2(\alpha - \beta)^2} > \frac{\sigma^2 + 2\mu^2}{2\sigma^2},$$

which can simplify to

$$\alpha^2 - (2\beta + \mu^2)\alpha + \beta^2 - \mu^2\beta > 0. \tag{19}$$

Clearly, (19) can be regarded as a quadratic inequality in $\alpha$, with its discriminant:

$$\Delta = \mu^4 + 8\mu^2\beta > 0.$$

This inequality implies that the inequality (19) holds when $\alpha \in (-\infty, \alpha_1) \cup (\alpha_2, +\infty)$, where

$$\alpha_1 = \beta + \frac{\mu^2 - \sqrt{\mu^4 + 8\mu^2\beta}}{2}$$

and

$$\alpha_2 = \beta + \frac{\mu^2 + \sqrt{\mu^4 + 8\mu^2\beta}}{2}. \tag{20}$$

In (20), the term $\mu^2 + \sqrt{\mu^4 + 8\mu^2\beta} > 0$, implying $\alpha_2 > \beta$. In contrast, we will derive $\alpha < \beta$ by the probability functions shown in Equations (16) and (17), particularly by the increasing property of $\Phi(\cdot)$. By this contradiction, we can say that $D_t > D$ holds only under the case of $\alpha \in (-\infty, \alpha_1)$, namely

$$\beta - \alpha + \frac{\mu^2 - \sqrt{\mu^4 + 8\mu^2\beta}}{2} > 0.$$

## B  OTHER RESULTS

### B.1  NUMERICAL ANALYSIS

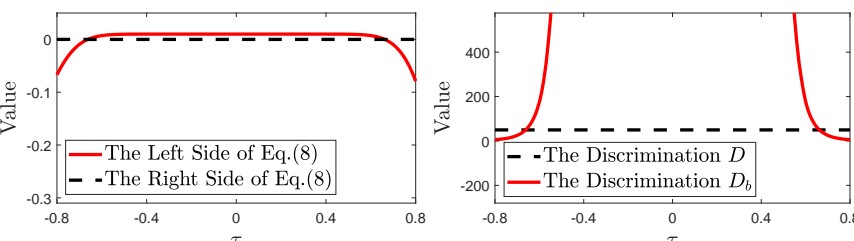

(a) Theoretical results for the data with distribution parameters $\mu = 0.99$ and $\sigma^2 = 0.02$
(b) Numerical results for the data with distribution parameters $\mu = 0.99$ and $\sigma^2 = 0.02$

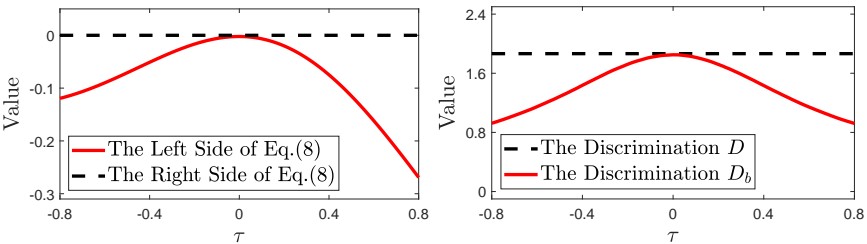

(c) Theoretical results for the data with distribution parameters $\mu = 0.76$ and $\sigma^2 = 0.42$
(d) Numerical results for the data with distribution parameters $\mu = 0.76$ and $\sigma^2 = 0.42$

Figure 7: Consider the binary quantization on two classes of data $X \sim N(\mu, \sigma^2)$ and $Y \sim N(-\mu, \sigma^2)$ as specified in Property 1. For two kinds of data with distribution parameters ($\mu = 0.99$, $\sigma^2 = 0.02$) and ($\mu = 0.76$, $\sigma^2 = 0.42$), the values for the left and right side of Equations (8) are provided in (a) and (c) respectively; and their discrimination $D$ and $D_b$ statistically estimated with Equations (5) and (6) are illustrated in (b) and (d), respectively.

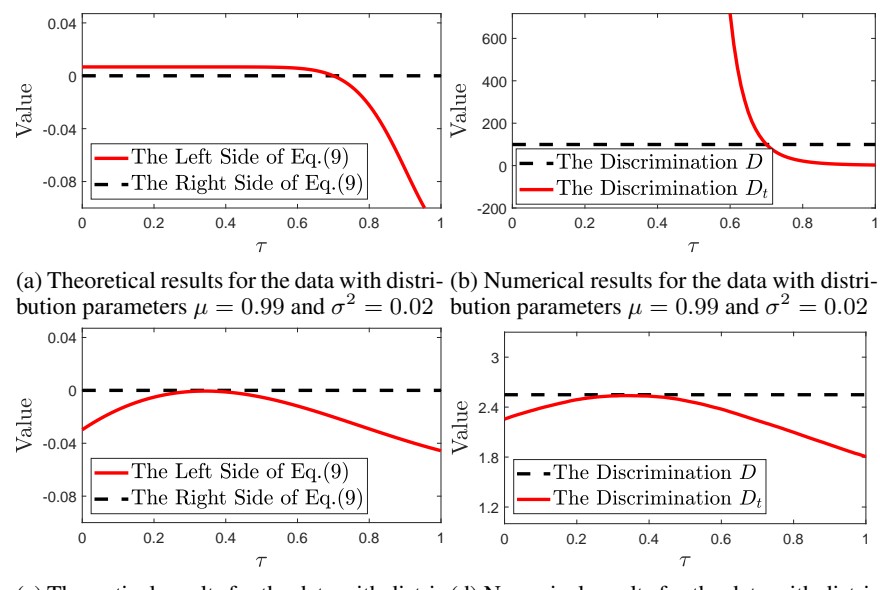

(a) Theoretical results for the data with distribution parameters $\mu = 0.99$ and $\sigma^2 = 0.02$

(b) Numerical results for the data with distribution parameters $\mu = 0.99$ and $\sigma^2 = 0.02$

(c) Theoretical results for the data with distribution parameters $\mu = 0.66$ and $\sigma^2 = 0.56$

(d) Numerical results for the data with distribution parameters $\mu = 0.66$ and $\sigma^2 = 0.56$

Figure 8: Consider the ternary quantization on two classes of data $X \sim N(\mu, \sigma^2)$ and $Y \sim N(-\mu, \sigma^2)$ as specified in Property 1. For two kinds of data with distribution parameters ($\mu = 0.99$, $\sigma^2 = 0.02$) and ($\mu = 0.66$, $\sigma^2 = 0.56$), the values for the left and right side of Equations (9) are provided in (a) and (c) respectively; and their discrimination $D$ and $D_t$ statistically estimated with Equations (5) and (7) are illustrated in (b) and (d), respectively.

## B.2 SYNTHETIC DATA

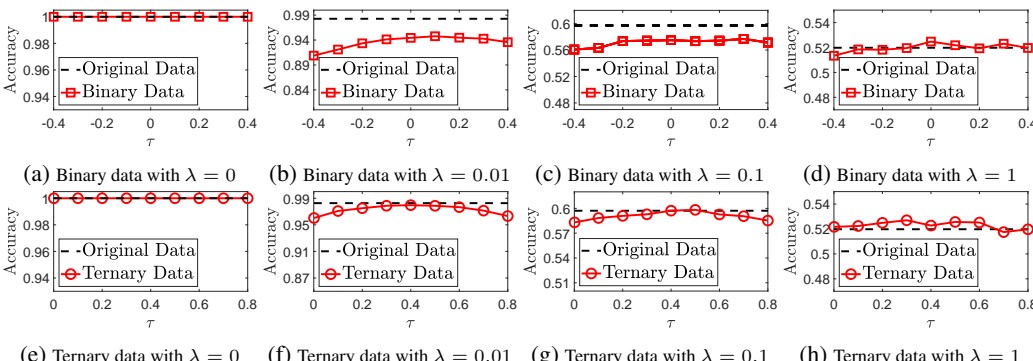

Figure 9: KNN (Cosine) classification accuracy for the 10,000-dimensional binary, ternary, and original data that are generated with the varying parameter $\lambda \in \{0, 0.01, 0.1, 1\}$, which controls the data sparsity.

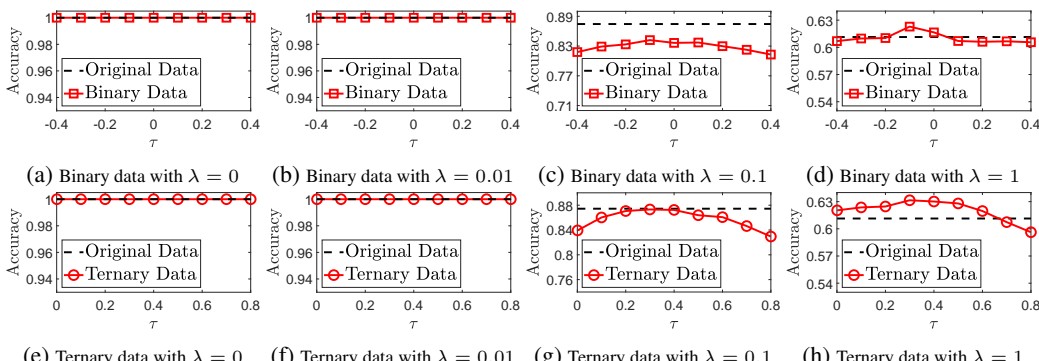

Figure 10: SVM classification accuracy for the 10,000-dimensional binary, ternary, and original data that are generated with the varying parameter $\lambda \in \{0, 0.01, 0.1, 1\}$, which controls the data sparsity.

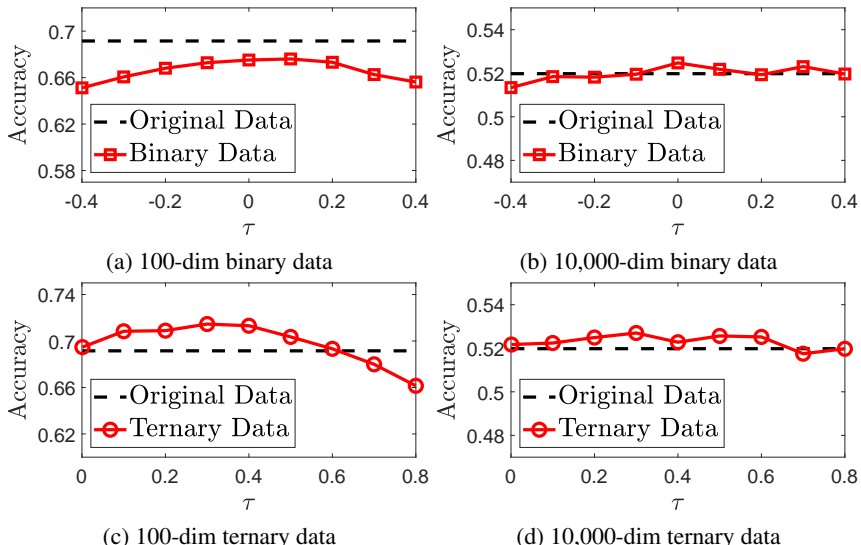

Figure 11: KNN (Cosine) classification accuracy for the binary, ternary, and original data generated with the sparsity parameter $\lambda = 1$, and with varying dimensions $n \in \{100, 10000\}$.

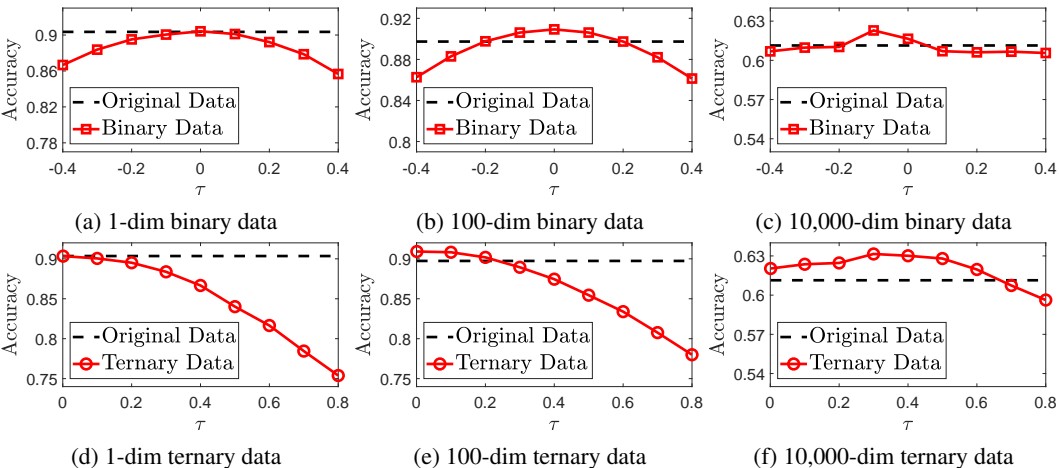

Figure 12: SVM classification accuracy for the binary, ternary, and original data generated with the sparsity parameter $\lambda = 1$, and with varying dimensions $n \in \{1, 100, 10000\}$.

## B.3 REAL DATA

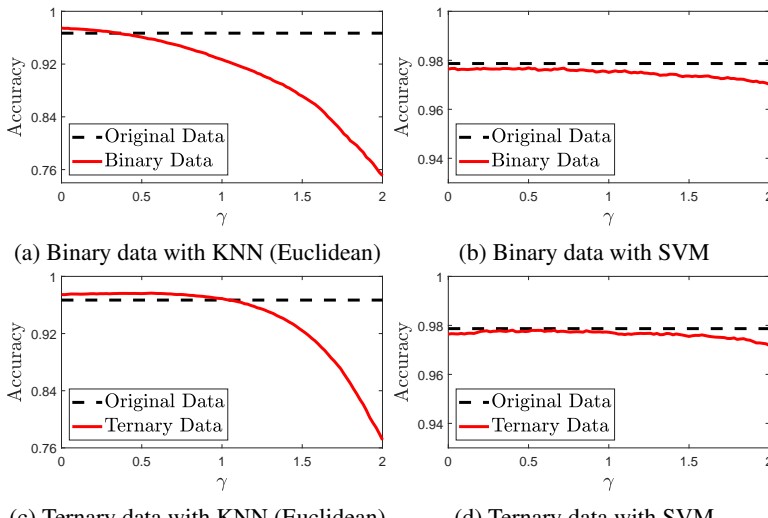

(a) Binary data with KNN (Euclidean)  (b) Binary data with SVM

(c) Ternary data with KNN (Euclidean)  (d) Ternary data with SVM

Figure 13: Classification accuracy for the binary, ternary, and original data by KNN (Euclidean distance) and SVM on CIFAR10. The parameter $\gamma$ corresponds to a threshold $\tau = \gamma \cdot \eta$, where $\eta$ denotes the average magnitude of the feature elements in all feature vectors.

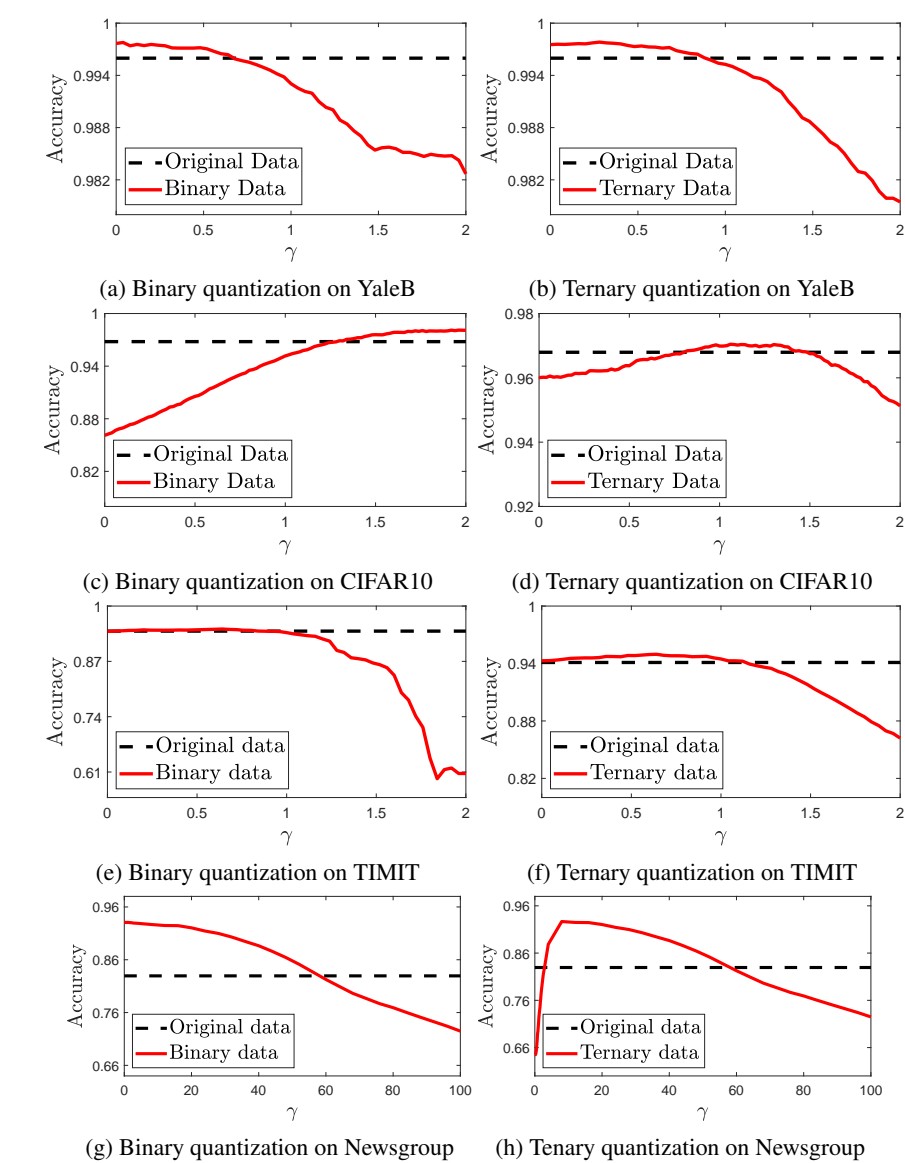

Figure 14: Classification accuracy for the binary, ternary, and original data by KNN (Cosine distance) on four different datasets. The parameter $\gamma$ corresponds to a quantization threshold $\tau = \gamma \cdot \eta$, where $\eta$ denotes the average magnitude of the feature elements in all feature vectors.

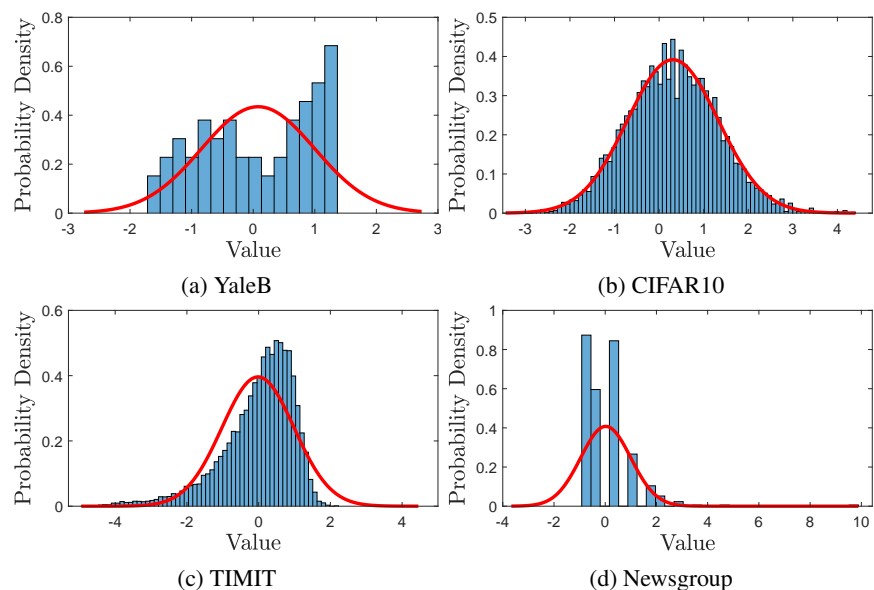

Figure 15: The histogram (blue bar) of the element values on one coordinate of the feature vectors within a single class of samples selected from four different datasets, accompanied with a Gaussian fitting curve (red line).

## C  RESPONSE TO REVIEWS

### C.1  CLASSIFICATION ACCURACY VS. FEATURE DISCRIMINATION VS. QUANTIZATION ERROR

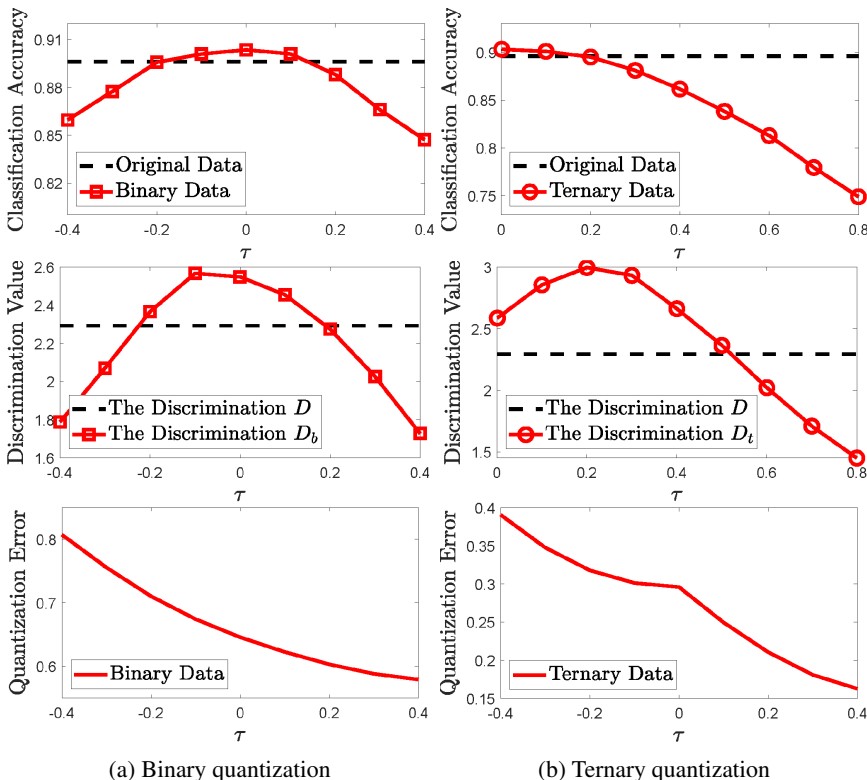

Figure 16: KNN (Euclidean distance) classification accuracy for the binary, ternary, and original synthetic data which are generated with the parameter $\lambda = 1$, and with data dimension equal to 1. For comparison, the feature discrimination values and quantization errors across different thresholds $\tau$ are provided for both binary and ternary data. Comments: It can be observed that the changing trend of classification values across $\tau$ can be reasonably reflected by feature discrimination, rather than by quantization errors.

## C.2  THE DATA DISTRIBUTION PARAMETER $\mu$ ESTIMATED WITH REAL DATA

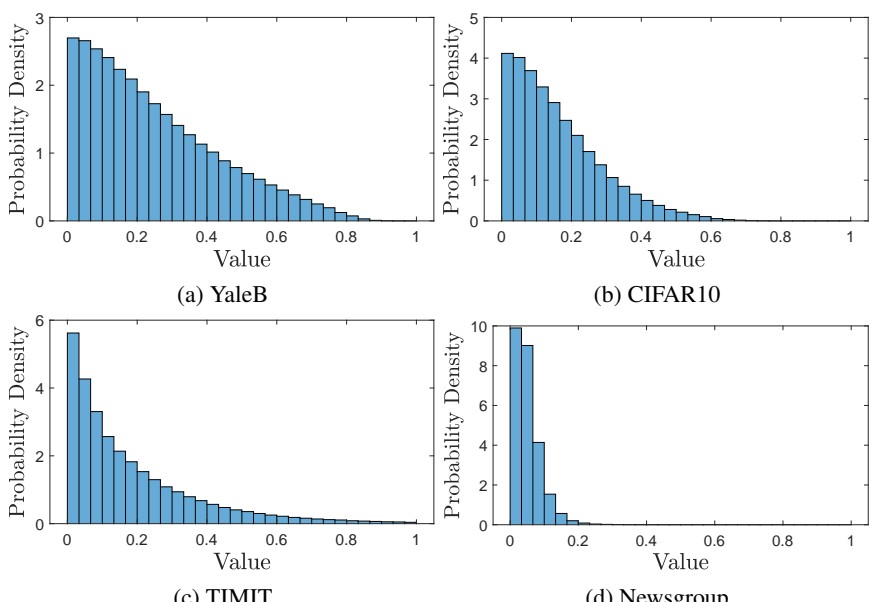

Figure 17: The histogram of the data distribution parameter $\mu$ (defined in Property 1) for each element (coordinate) of the feature vectors used in binary classification. Comments: It can be seen that with certain probabilities, the $\mu$ value of each feature element will fall within the regions of $(0.76, 1)$ and $(0.66, 1)$, which supports achieving improved classification by binary and ternary quantization. Despite the fact the the probabilities are not large, namely the amount of feature elements falling within $(0.76, 1)$ or $(0.66, 1)$ is relatively few, as widely proved in our experiments, we can still obtain the desired thresholds $\tau$ that support improving classification on these real data. This robustness should be attributed to the fact that classification performance is mainly determined by a few important feature elements of large magnitude, such as the ones with absolute means $\mu$ falling within $(0.76, 1)$ or $(0.66, 1)$.

## C.3 THE BINARY ATTRIBUTE OF FEATURE ELEMENTS ACROSS MULTIPLE CLASSES

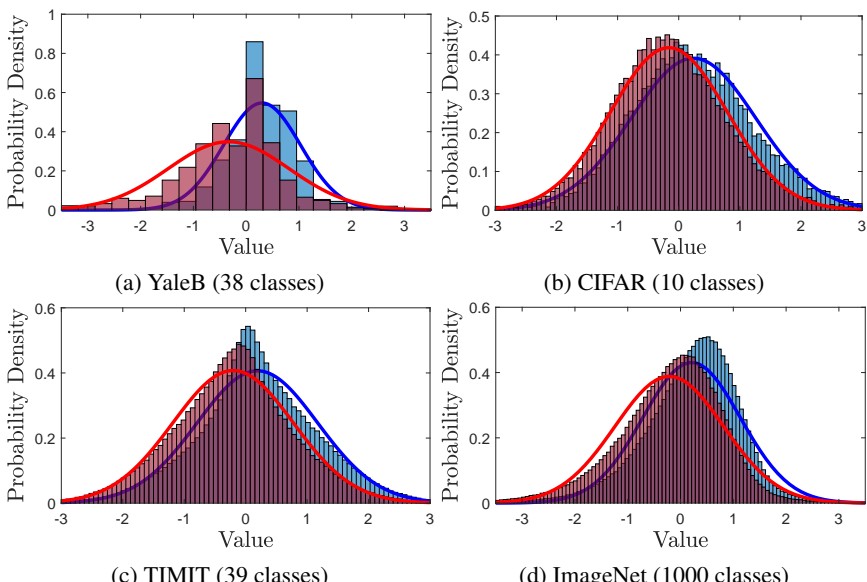

(a) YaleB (38 classes)

(b) CIFAR (10 classes)

(c) TIMIT (39 classes)

(d) ImageNet (1000 classes)

Figure 18: Two histograms are drawn, one for the feature elements with values less than zero (dark red) and the other for those greater than zero (dark blue). The feature elements are collected from a common coordinate of feature vectors across all classes. Both histograms are fitted with Gaussian curves. Comments: It can be seen that both histograms approximately exhibit Gaussian distributions, with their two means separable. This indicates the binary nature (strong and weak) of the feature elements at each coordinate of feature vectors, regardless of the number of classes from which the feature vectors are drawn. This property allows us to generalize our binary classification-based feature discrimination analysis to multiclass classification scenarios. The reason is as follows. Consider a feature vector $\mathbf{x} = [x_1, x_2, .., x_n]^\top$ for a given sample, where each element $x_i$ corresponds to a specific feature attribute, such as frequencies in DCT features, scale and spatial positions in DWT features, or filters in convolutional features. The value of $x_i$ indicates the strength of the $i$-th attribute present within the sample. The strength of $x_i$ can characterized by two distinct states: strong and weak, which reflect the presence or absence of the $i$-th attribute in the sample. The two states are evidenced in our statistical analysis of the $x_i$ values in real-data feature vectors $\mathbf{x}$. The results, depicted in this figure, show that the large (>0) and small values (<0) both exhibit Gaussian distributions, with the means of theses distributions representing the strong and weak states, respectively. Given this understanding, the classification of each attribute (coordinate) $x_i$ in feature vectors $\mathbf{x}$ can be considered a binary classification problem, regardless of the number of classes from which the feature vectors $\mathbf{x}$ are drawn. Consequently, we can conclude that the capability of quantization to improve binary classification can also be extended to multiclass classification, provided that the Gaussian distributions of the two attributes at each coordinate of feature vectors are sufficiently separable, as required in Theorems 1 and 2.

### C.4 Nonlinear Classifiers: MLP and Decision Trees

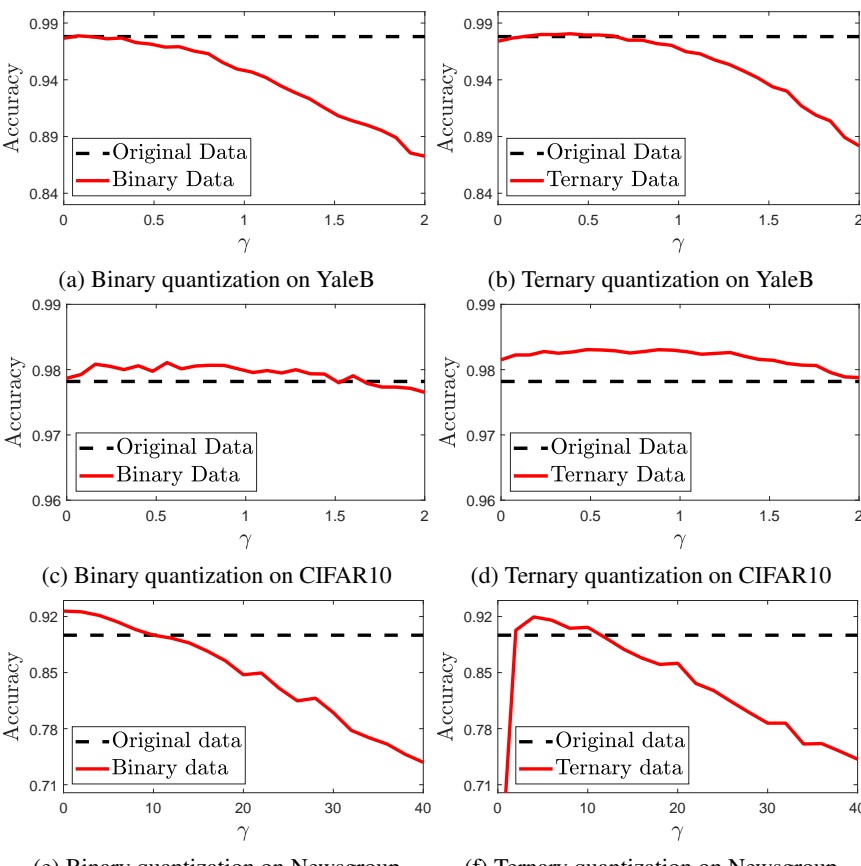

(a) Binary quantization on YaleB

(b) Ternary quantization on YaleB

(c) Binary quantization on CIFAR10

(d) Ternary quantization on CIFAR10

(e) Binary quantization on Newsgroup

(f) Ternary quantization on Newsgroup

Figure 19: MLP-based binary classification accuracy for the binary, ternary, and original data on three different datasets. The parameter $\gamma$ corresponds to a quantization threshold $\tau = \gamma \cdot \eta$, where $\eta$ denotes the average magnitude of the feature elements in all feature vectors. Comments: Despite the fact that our linear feature discrimination analysis on quantized data may not directly extend to nonlinear classification scenarios, experiments using classifiers MLP and decision trees demonstrate that binary and ternary quantization can achieve improved or at least comparable classification results even with nonlinear classifiers. This should be attributed to the fact that nonlinear classifiers generally involve fundamental linear operations, that evaluate the linear discrimination among features or model parameters.

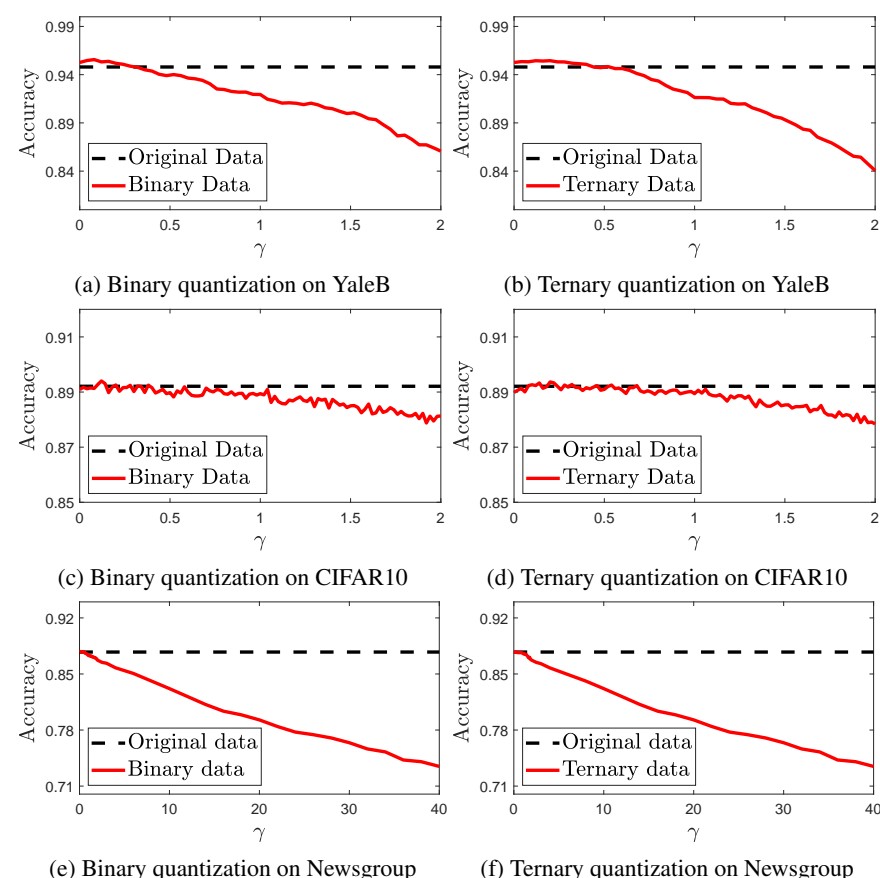

(a) Binary quantization on YaleB

(b) Ternary quantization on YaleB

(c) Binary quantization on CIFAR10

(d) Ternary quantization on CIFAR10

(e) Binary quantization on Newsgroup

(f) Ternary quantization on Newsgroup

Figure 20: Decision trees-based binary classification accuracy for the binary, ternary, and original data on three different datasets. The parameter $\gamma$ corresponds to a quantization threshold $\tau = \gamma \cdot \eta$, where $\eta$ denotes the average magnitude of the feature elements in all feature vectors. Comments: Despite the fact that our linear feature discrimination analysis on quantized data may not directly extend to nonlinear classification scenarios, experiments using classifiers MLP and decision trees demonstrate that binary and ternary quantization can achieve improved or at least comparable classification results even with nonlinear classifiers. This should be attributed to the fact that nonlinear classifiers generally involve fundamental linear operations, that evaluate the linear discrimination among features or model parameters.

## C.5 Binary and Multiclass Classifications on ImageNet1000

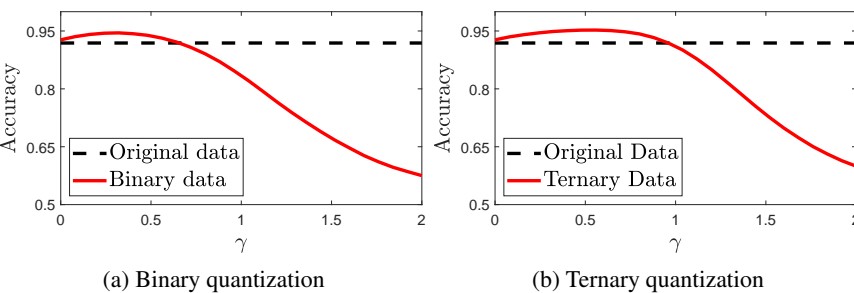

(a) Binary quantization  (b) Ternary quantization

Figure 21: Binary classification accuracy for the binary, ternary, and original data in ImageNet1000, using the classifier KNN (Euclidean distance). The parameter $\gamma$ corresponds to a quantization threshold $\tau = \gamma \cdot \eta$, where $\eta$ denotes the average magnitude of the feature elements in all feature vectors. Comments: It is evident that there are quantization thresholds $\tau$ that can improve the binary classification accuracy of ImageNet1000. Given the complexity of ImageNet1000, this validates the generalizability of our findings.

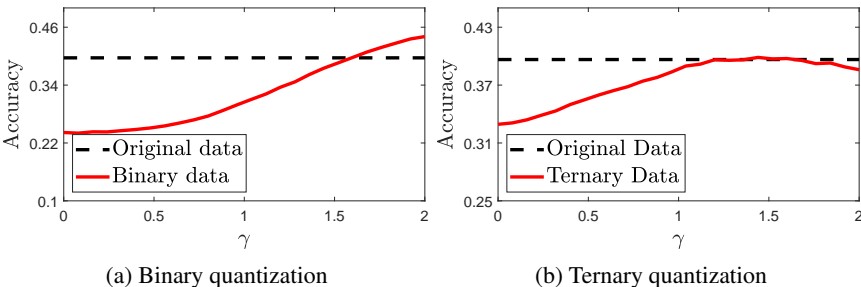

(a) Binary quantization  (b) Ternary quantization

Figure 22: Multiclass classification accuracy for the binary, ternary, and original data in ImageNet1000, using the classifier KNN (Euclidean distance). The parameter $\gamma$ corresponds to a quantization threshold $\tau = \gamma \cdot \eta$, where $\eta$ denotes the average magnitude of the feature elements in all feature vectors. Comments: It can be seen that there are quantization thresholds $\tau$ that can improve the multiclass classification accuracy of ImageNet1000. This validates that our feature discrimination analysis, rooted in binary classification, can be extended to multiclass classification, owing to the binary state of the feature elements sharing a common coordinate across different classes. See Figure 18 for detailed explanations.

