# OpenReview forum: "Feature Discrimination Analysis for Binary and Ternary Quantization"
_ICLR.cc/2025/Conference — Submitted to ICLR 2025_

### Official Review · Reviewer_JS4C · 2024-10-31

**Soundness:** 2
**Presentation:** 3
**Contribution:** 2
**Rating:** 5
**Confidence:** 4

**Summary:**

This paper proposes a method to analyze the impact of binary and ternary quantization on the performance of classification tasks by focusing on feature discrimination rather than quantization errors. Unlike traditional approaches that primarily uses quantization errors to estimate performance degradation, this work demonstrates that by selecting a proper quantization threshold, binary and ternary quantization can sometimes improve classification accuracy by enhancing feature discrimination. Through theoretical analysis and empirical experiments on synthetic and real datasets, the paper provides valuable insights into how specific quantization thresholds can yield optimal classification performance.

**Strengths:**

1. The paper analyzes the impact of binary and ternary quantization on classification performance based on feature discrimination, which directly correlates to classification performance and offers an alternative to quantization error as a metric.

2. Theoretical derivations are well-supported with numerical experiments across different types of datasets, including synthetic and real datasets.

**Weaknesses:**

1. No large datasets were used: The datasets used in classification tasks are too small. Experiments on large datasets are needed to verify whether the conclusions and findings of this paper are still valid.

2. The classification tasks are too simple: The authors verified the impact of quantization on feature discrimination only in binary classification tasks which are too simple and the conclusions and findings of this paper may not work for complex classification tasks.

3. Limited classifiers were studied: The work only studied the impact of binary and ternary quantization on feature discrimination of KNN and SVM. How about MLP or decision trees or other classifiers?

**Questions:**

1. This paper said the quantization on the data with large sparsity will have a negative effect on the performance which is contradictory to a current paper [1].
[1] Chen, M., & Li, W. (2023). Ternary and Binary Quantization for Improved Classification. Authorea Preprints.

2. Is the proposed feature discrimination-based quantization analysis approach applicable to quantization methods beyond binary and ternary?

---

> ### Author Response · Authors · 2024-11-23
> **Response to Reviewer JS4C**
>
> The authors would like to thank the reviewers for sparing your precious time to review our manuscript.  **We have  addressed all reviewer concerns by providing a plethora of experiments and detailed explanations in the Appendix C, Figures 16-22**. We will answer the questions in the order they were raised by the reviewers.
>
> **Q1:** No large datasets were used.
>
> **A1:** Thank you. Following the suggestion, we have provided the **binary** and **multiclass** classification results for the large dataset ImageNet1000 in **Figures 21 and 22**. The results demonstrate the existence of quantization thresholds that can enhance classification compared to the original non-quantized data. Moreover, it is worth mentioning that the advantage of quantization in improving the **multiclass** classification of ImageNet is also confirmed in the empirical study in [r].
>
> [r] Weizhi Lu, Mingrui Chen, Kai Guo, and Weiyu Li. Quantization: Is it possible to improve classification? In Data Compression Conference, pp. 318–327. IEEE, 2023.
>
>
> **Q2:** The classification tasks are too simple: The authors verified the impact of quantization on feature discrimination only in binary classification tasks which are too simple and the conclusions and findings of this paper may not work for complex classification tasks.
>
> **A2:** Thank you. We have to emphasize that  establishing our feature discrimination analysis on **binary classification** is   **necessary** and **rational**. There are two major reasons. 1) Firstly, the accuracy of binary classification can directly reflect the discrimination between two classes of data. 2) Secondly, binary classification analysis is fundamental in machine learning and serves as a foundational concept that can be extended to the study of multi-class classification. For more  evidences and analyses, please see A1,  A3 and A4, as well as the  A4 to the Reviewer **aSct**.
>
>
>
>
>
> **Q3:** How about the classification with MLP or decision trees or other classifiers?
>
> **A3:** Thank you. As previously mentioned, we have utilized linear classifiers such as KNN and SVM to directly assess the linear feature discrimination that we have theoretically estimated between two classes of data. In contrast, other nonlinear classifiers like decision trees and MLP typically involve feature selection operations and may not directly capture the linear feature discrimination between two classes of data. Therefore, it is not appropriate to evaluate our linear feature discrimination analysis using nonlinear classifiers.
>
> Following the reviewer’s suggestion, we have provided the classification results using MLP and decision trees in the **Figures 19 and 20**. Interestingly, both classifiers exhibit enhanced classification performance after binary and ternary quantization. The improved classification can be attributed to the fact that nonlinear classifiers generally involve fundamental linear operations, that evaluate the linear discrimination between features or model parameters.
>
> **Q4:** This paper said the quantization on the data with large sparsity will have a negative effect on the performance which is contradictory to a current paper [1]. [1] Chen, M., & Li, W. (2023). Ternary and Binary Quantization for Improved Classification. Authorea Preprints.
>
> **A4:** Thank you. The formal publication [r] of the reference mentioned by the reviewer has been discussed in our manuscript. Our results are not contradictory. In the empirical study of [r], it is observed that the binary and ternary quantization on some commonly-used sparse features, like the DWT of YaleB, the convolution features of Cifar10 and ImageNet, tends to improve classification. Similar results on these sparse features are also noted in our experiments. In our study, we further observe that when the feature vector becomes *overly* sparse, namely containing too many elements with small-magnitude means $|\mu_i|$, the advantage of quantization will diminish. This is because small $|\mu_i|$ values are not conducive to enhancing feature discrimination through quantization, according to our theoretical and numerical analysis.
>
> [r] Weizhi Lu, Mingrui Chen, Kai Guo, and Weiyu Li. Quantization: Is it possible to improve classification? In Data Compression Conference, pp. 318–327. IEEE, 2023.
>
> **Q5:** Is the proposed feature discrimination-based quantization analysis approach applicable to quantization methods beyond binary and ternary?
>
> **A5:** Thank you. Technically, our feature discrimination analysis on binary and ternary quantization can be expanded to cope with other quantization methods with larger bit-widths. As the bit-width increases, the value range of quantized data expands, significantly elevating the analytical complexity of the feature discrimination function. To mitigate this complexity, approximation methods may be necessary. This work will be left for our future research.

---

> > ### Comment · Reviewer_JS4C · 2024-11-26
> >
> > Thank you for the authors’ response. I appreciate their efforts in enhancing the manuscript by validating other classifiers after binary and ternary quantization, along with providing numerical results on large datasets. However, I still have reservations about whether the proposed feature discrimination-based quantization analysis approach can effectively enhance classification performance for large deep models after binary and ternary quantization. If the authors can provide additional experimental evidence demonstrating the method’s impact on large deep models, I would be more inclined to raise the score.

---

> ### Author Response · Authors · 2024-11-26
> **Response to Reviewer JS4C:  The analysis of binary/ternary quantization on deep networks**
>
> Dear Reviewer JS4C,
>
> Thank you for kindly considering our response. **Recently, binary and ternary quantization methods have achieved outstanding performance in the quantization of deep networks [r1,r2].** For instance, [r3] found that the two quantization methods can **improve** the classification accuracy of deep networks on relatively small datasets, such as MNIST, CIFAR-10, and SVHN. Similarly, on the larger ImageNet dataset, [r4] reported an **improved** performance. Generally, despite suffering from significant quantization errors, most quantized networks can still match or exceed the performance of their full-precision counterparts [r1,r2].
>
> [r1] Amir Gholami, Sehoon Kim, Zhen Dong, Zhewei Yao, Michael W Mahoney, and Kurt Keutzer. A survey of quantization methods for efficient neural network inference. In Low-Power Computer Vision, pp. 291–326. Chapman and Hall/CRC, 2022.
>
> [r2] Haotong Qin, Ruihao Gong, Xianglong Liu, Xiao Bai, Jingkuan Song, and Nicu Sebe. Binary neural networks: A survey. Pattern Recognition, 105:107281, 2020.
>
>
> [r3] Zhouhan Lin, Matthieu Courbariaux, Roland Memisevic, and Yoshua Bengio. Neural networks with few multiplications. In International Conference on Learning Representations, 2016b.
>
> [r4] Zhu C, Han S, Mao H, et al. Trained ternary quantization[J]. Arxiv preprint arxiv:1612.01064, 2016.
>
>
>
> Given the significant quantization errors of binary/ternary quantization, it is apparent that the superior performance of these quantization methods in deep networks can hardly be explained by quantization errors. **Instead, this superior performance can be reasonably explained though our linear feature discrimination analysis, as deep networks fundamentally comprise simple, linear operations. Specifically, the convolution between each filter and feature patch represents a linear operation, which measures the linear discrimination between the filter and the patch.** Empirically, when applied to relatively small datasets as previously mentioned, quantization methods are more prone to achieving improved or at least comparable classification performance compared to non-quantized networks. This is because the feature patches in these simple datasets have relatively high discrimination, leading to high aggregation and separability in their distributions. These distributions, in turn, align well with the Gaussian distribution assumption underlying our theoretical analysis. In contrast, this distribution assumption may be difficult to satisfy for more complex data, such as ImageNet. In this case, we need to carefully design the network structure, by adjusting the size and number of filters in each layer, in terms of the discrimination of the feature patches in that layer. By employing the quantization method introduced in the paper, it should be possible to achieve superior quantized networks. This is the work we are currently undertaking, and we have achieved some advancements.
>
> Finally, we would like to emphasize that **our feature discrimination-based quantization analysis offers a **fundamental** and **significant contribution** to  the field of data quantization.** It is the first to theoretically prove that quantization can improve, rather than degrade, the performance of data classification. This challenges the traditional belief that larger quantization errors generally lead to lower classification performance, **establishing a theoretical foundation for developing better quantization methods.**
>
> Once again, we deeply appreciate the time and effort you have dedicated to reviewing our manuscript.
>
> Best regards,
>
> The authors

---

### Official Review · Reviewer_Ez39 · 2024-11-01

**Soundness:** 2
**Presentation:** 3
**Contribution:** 2
**Rating:** 5
**Confidence:** 3

**Summary:**

This paper presents the study of binary and ternary quantization for the classification problem through the feature discrimination capability analysis. The main contribution of this paper is to prove that quantization errors do not necessarily lead to decreased classification performance. The proof is done through theoretical analysis and experiments with simulated and real-life data sets. Thus, the estimation of the classification performance can be achieved by examining the feature discrimination of quantized data rather than only the quantization errors.

**Strengths:**

As the authors claim, this may be the first study to exploit feature discrimination to analyze the impact of quantization on classification. One important finding is that the quantization thresholds have an impact on feature discrimination and classification performance. The authors conducted theoretical analyses to prove that binary and ternary quantization can enhance feature discrimination between two classes of data. The choice of the quantization threshold becomes a key factor for better classification performance.

The work is original and interesting, and the paper is well written and presented. The idea was proved through numerical analysis and experiments with simulated and real-time data.

**Weaknesses:**

Although the paper is easy to follow, some problems still need to be clarified.

-  As stated in 2.4, the major goal of this paper is to investigate whether there exist threshold values in binary and ternary quantization to improve feature discrimination. This is confusing, as different threshold values will result in different feature discrimination measures. Then, what is the significance of this finding? Could you please clarify the practical implications of finding threshold values that improve feature discrimination or how the finding will help optimize the classification process?

- The abstract mentions classification generally but does not specify the number of classes. In the study, the experiment is binary classification, even for real-life data. Does that imply any relation between binary and ternary quantization with binary classification or even ternary classification? This raises the question: is this finding for general classification or binary classification only? What additional work would be needed to generalize the findings to multi-class (>3) classification problems?

- The design of the experiments can be improved to support the claims directly. For instance, the experiments whose results are presented in 4.1.2 considered data sparsity, data dimension, different classifiers, and the difference between binary and ternary quantization. To some extent, the results raise more questions. The comparison between binary and ternary quantization does not derive any solid conclusion, just stating, "yield superior performance." The variables considered in the experiments are not directly related to the core topic. The variables should include the "feature discrimination measure" and "quantization error," and experiments should consider the three scenarios with original data, binary, and ternary quantization.

- In Figure 3, the classification with binary or ternary quantization does not always achieve a better result. Then, an optimal threshold value is expected, but how? This is not available in this study. In practice, how can this value be determined for varied scenarios (such as data dimension)?

- The paper criticizes using quantization errors to estimate classification performance. Is it possible to show the quantization errors in the experiments as a baseline? This will help better understand the value of the work. A comparison of quantization errors and classification performance across different threshold values, to directly illustrate the limitations of using quantization errors as a proxy for classification performance may be beneficial.

**Questions:**

The experiment results show classification accuracies with original, binary, and ternary data. Is it possible to show the feature discrimination capability (e.g., the ratio between inter-class and intra-class scatters) represented with some numbers and quantization errors? This is what the study directly deals with.

From the experimental results, we may conclude that classification with binary or ternary quantization does not always achieve a better result, even for binary classification. An optimal threshold value is essential, but there is still no solution to find that.

Finally, it would be better to clarify whether the finding is for general classification or binary classification and why.

---

> ### Author Response · Authors · 2024-11-23
> **Response to Reviewer Ez39**
>
> The authors would like to thank the reviewers for sparing your precious time to review our manuscript.  **We have  addressed all reviewer concerns by providing a plethora of experiments and detailed explanations in the Appendix C, Figures 16-22**. We will answer the questions in the order they were raised by the reviewers.
>
>
>
> **Q1:**  The significance of this finding, the practical implications of finding threshold values that improve feature discrimination, how the finding will help optimize the classification process?
>
> **A1:** Thank you. As commented by the reviewer, the feature discrimination measure is a function of the quantization threshold $\tau$, which tends to vary with $\tau$. In Theorems 1 and 2, we prove that the feature discrimination of quantized data is higher than the one of original data, if the quantization threshold $\tau$ satisfies the inequalities (8) and (9).  As discussed in the A4 to Reviewer **aSct**,  the desired $\tau$ can be estimated with approximate solution algorithms, like the bisection method.
>
>
> With the method described above for identifying the threshold $\tau$ that improves feature discrimination, we can broadly apply it in current binary or ternary quantization tasks, such as large-scale retrieval  and deep network quantization, in order to achieve better quantization/classification performance.
>
>
>
>
> **Q2:** Is this finding for general classification or binary classification only? What additional work would be needed to generalize the findings to multi-class (>3) classification problems?
>
> **A2:** Thank you. It is noteworthy that our findings are applicable to both binary and multiclass classifications, both theoretically and empirically. For details, please see our response A4 to the previous Reviewer **aSct**.
>
>
> **Q3:** The variables considered in the experiments of Section 4.1.2 should include the "feature discrimination measure" and "quantization error," and experiments should consider the three scenarios with original data, binary, and ternary quantization.
>
> **A3:**  Thank you. In Section 4.1, we conduct classification on synthetic data with two main purposes. Firstly, we aim to further validate some results obtained from our previous theoretical and numerical analyses. For example, in comparing binary and ternary quantization, our goal is to demonstrate that ternary quantization can encompass a *wider* range of $\tau$ values leading to improved classification. Secondly, we use synthetic data to simulate real-world data with varying sparsity levels and dimensions to evaluate the robustness and generalizability of our theoretical findings.
>
> Following the reviewer’s suggestion, in **Figure 16** we have compared the changing trends of classification accuracy, feature discrimination and quantization errors across different threshold values $\tau$. It can be seen that the classification performance can be reasonably reflected by feature discrimination, rather than by quantization errors.
>
>
>
> **Q4:** In Figure 3, the classification with binary or ternary quantization does not always achieve a better result. Then, an optimal threshold value is expected, but how?
>
> **A4:** Thank you. Note that our objective is to demonstrate the existence of the thresholds $\tau$ that can enhance classification accuracy. The desired thresholds have been identified across all results in Figure 3, and are consistently observed in the majority of classification experiments conducted on synthetic and real datasets. In a few instances, we fail to derive the desired thresholds, primarily due to two key reasons. 1) Firstly, the data distributions do not align with our theoretical prerequisites: the data distributions between two classes should be sufficiently separable. 2) Secondly, it is essential to note that our theoretical framework relies on the Euclidean distance metric for similarity assessment. To assess the generalizability of our findings, we also evaluate classification using another commonly used metric--the **correlation (cosine)** distance in KNN, as illustrated in Figures 10 and 12. A few results fail to provide a threshold that improves classification, attributed to the fact that the correlation metric is not as effective as the Euclidean distance in capturing the similarity between binary/ternary data, particularly when quantifying the distance between 0 and 1/-1.
>
> Regarding how to derive the quantization threshold $\tau$ that enhances feature discrimination, please see the A4 to Reviewer **aSct**.
>
>
>
> **Q5:**  A comparison of quantization errors and classification performance across different threshold values, to directly illustrate the limitations of using quantization errors as a proxy for classification performance may be beneficial.
>
> **A5:** Thank you. This question has been answered in A3.
>
> **Q6:** The three questions raised in the “Questions” part.
>
> **A6:** The three questions are similar to the previous ones, and can find their answers in A3, A4 and A2, respectively.

---

> ### Author Response · Authors · 2024-11-29
> **Response to Reviewer Ez39:  Multiclass classification and  linear-and-nonlinear classifiers based classification**
>
> **Q1:** It is not clear how many classes were considered in the multi-class case. Will the number of classes impact the results?
>
> **A1:** Thank you. To validate the wide applicability of our theoretical discoveries, we have conducted a challenging **1000-class** classification on ImageNet1000, as shown in Figure 22. Further evidence supporting our findings can be found in [r], where multiclass classification tasks were performed on datasets such as YaleB (38 classes), FashionMNIST (10 classes), and CIFAR10 (10 classes).
>
> [r] Weizhi Lu, Mingrui Chen, Kai Guo, and Weiyu Li. Quantization: Is it possible to improve classification? In Data Compression Conference, pp. 318–327. IEEE, 2023.
>
> As analyzed in the final paragraph of Section 4.2.2 and in the comment of Figure 18, **our finding on binary classification can be extended to multiclass classification, when feature elements sharing identical coordinates across diverse classes exhibit a binary state, with each state satisfying a Gaussian distribution.** This characteristic hinges on the complexity of the distribution of feature elements at the same coordinates. Empirical evidence suggests that this property generally holds true for typical features across various datasets, including YaleB (38 classes), CIFAR10 (10 classes), TIMIT (39 classes), and ImageNet (1000 classes), as illustrated in Figure 18.
>
> **Q2:** Another observation is that the results depend on the varied classification methods, e.g., KNN, decision tree, or SVM. Thus, binary or ternary quantization is less significant than changing different classification methods. Thus, I may want to keep or lower my scores.
>
> **A2:** Thank you. It is necessary to underscore our major contribution: we are the first to suggest the use of linear discrimination, as opposed to focusing on quantization errors, to investigate the impact of quantization on data classification. Furthermore, we have theoretically demonstrated that extremely low bitwidth binary and ternary quantization can improve  classification accuracy for the original data, contrary to the common belief that it may decrease accuracy.
>
> Theoretically, our  theoretical findings can  directly apply to linear, binary classification, as extensively validated by the KNN and SVM-based binary classification experiments provided in the paper. Empirically, our findings can also be extended to multiclass (Figure 22) and nonlinear (Figures 19 and 20) classifications, because both types  generally incorporate linear operations, making them amenable to investigation using our linear discrimination analysis.  For detailed explanations, the final paragraph of Section 4.2.2 and in the comments of Figure 18.
>
> Moreover, it is worth noting that while we have achieved satisfactory performance in **multiclass and nonlinear classifications**, conducting experiments in these areas is **not essential** for confirming our theoretical findings on the linear discrimination of quantized data.  These findings can be directly validated through linear, binary classification using KNN and SVM.
>
> **In summary, it can be said that our paper is comprehensive both theoretically and experimentally, making a fundamental and significant contribution to the field of data quantization.**

---

### Official Review · Reviewer_aSct · 2024-11-06

**Soundness:** 4
**Presentation:** 4
**Contribution:** 3
**Rating:** 8
**Confidence:** 4

**Summary:**

This paper proposes “feature discrimination” to analyze the impact of quantization on classification, which offers a more direct and rational assessment of classification performance rather than relying on quantization error as previous researches asses classification performance roughly.

**Strengths:**

1. The motivation is interesting and the addressed quantization analysis problem is meaningful.
2. The proposed feature discrimination analysis is pretty novel.
3. Sufficient and rigorous theoretical proof to derive the value range of the quantization threshold τ based on µ and σ for binary quantization and ternary quantization, respectively.
4. Clear method statement, careful logic and sufficient explanations.
5. Adequate experiments on both synthetic data and real data.

**Weaknesses:**

1. As for Eq. (5), further explanations are need to state why discrimination between two classes of data can be formulated to Eq. (5) for clarify.
2. In the Remarks paragraph on P4, the authors said “it is demonstrated that the desired thresholdτdoes exist, when the two classes of data X∼N(µ, σ2 ) and Y∼ N(−µ, σ2 ) are assigned appropriate values for µ and σ”. Are µ and σ in the quantization space set? I mean once the quantization method is used, the distribution in the quantization space is determinate. How can we guarantee appropriate values for µ and σ? In other words, if the quantization space does not meet the condition, is the analysis reasonable or applicative?
3. In real data experiments, we see the value ranges for the threshold τ, it is better to provide the values for µ and σ in the real data case to further analyze the influence of the distributions for quantization.

**Questions:**

1. What if it is not a binary classification problem (more than two classes) or for image classification problem with muti-labels?
2. The paper provides theoretical analysis that the appropriate threshold τ exists, but how to set it not depending on the classification accuracy?

---

> ### Author Response · Authors · 2024-11-23
> **Response to Reviewer aSct**
>
> The authors would like to thank the reviewers for sparing your precious time to review our manuscript. **We have  addressed all reviewer concerns by providing a plethora of experiments and detailed explanations in the Appendix C, Figures 16-22**. We will answer the questions in the order they were raised by the reviewers.
>
>
> **Q1:** As for Eq. (5), further explanations are need to state why discrimination between two classes of data can be formulated to Eq. (5) for clarify.
>
> **A1:** Thank you. Following the suggestion, we have elucidated the structure of Eq. (5) based on the conventional definition of linear discriminant analysis (LDA).
>
> **Q2:**  In practical terms, do the distribution parameters $\mu$ and $\sigma$ of standardized data as conditioned in our theoretical analysis actually exist?
>
> **A2:** Thank you. In our understanding, the reviewer means that our analysis is rooted in the standardized data $X∼N(\mu, \sigma^2 )$ and $X∼N(-\mu, \sigma^2 )$, while the analysis results/conditions (dependent on $\mu$ and $\\sigma$) may not align with the original data distributions, $X∼N(\mu_1, \sigma_1^2 )$ and $X∼N(\mu_2, \sigma_2^2 )$. However, this concern is unwarranted, given the explicit relationships between the data distribution parameters before and after standardization, as delineated in Eqs. (3) and (4). The rationale behind this assertion is further elaborated below.
>
> Firstly, it is noteworthy that our theoretical findings concerning Equations (8) and (9) are solely dependent on the parameter $\mu$ of standardized data, given that $\sigma^2=1-\mu^2$. By analyzing the conditions about Equations (8) and (9) in Section 3.2, it is demonstrated that the gain in feature discrimination can be achieved for binary and ternary quantization, when $\mu$  falls within the ranges of (0.76,1) and (0.66,1), respectively. By the relationship between original and standardized data, as illustrated in Equations (3) and (4), it can be seen that larger values of $\mu$ within the desired ranges of (0.76,1) and (0.66,1) can be attained from the original data, when the data have a substantial difference between the means ($\mu_1$-$\mu_2$) and a relatively small deviation $\sigma$. Put simply, as concluded in Section 3.2, quantization can yield feature discrimination gains when the original data exhibit sufficient separability.
>
>
> **Q3:**  In real data experiments, we see the value ranges for the threshold $\tau$, it is better to provide the values for $\mu$ and $\sigma$ in the real data case to further analyze the influence of the distributions for quantization.
>
> **A3:** Thank you. Following our earlier discussion in A2, it is known that the feature discrimination gain can be obtained for binary and ternary quantization, when $\mu$ falls within the ranges of $(0.76,1)$ and $(0.66,1)$, respectively. In **Figure 17**, we have depicted and analyzed the distributions of $\mu$ (across each data dimension) for real data. For more details, please see the comments of Figure 17.
>
>
>
> **Q4:**  What if it is not a binary classification problem (more than two classes) or for image classification problem with muti-labels?
>
> **A4:** Thank you. In our feature discrimination analysis on quantized data, **we choose to focus on linear, binary classification for two major reasons.** 1) Firstly,  linear binary classification can directly reflect the feature discrimination between two classes of data. 2) Secondly,  linear binary classification is a fundamental concept in machine learning. The insights gained from this analysis can be extended to multiclass classification.
>
> Recent research [r]  has found that binary and ternary quantization can improve  the accuracy of **multiclass** classification. Also, this property is observed in  our new experiments  provided  in **Figure 22**.  This implies that our theoretical findings on binary classification can be extended  to multiclass classification.  We have explained this problem **in the comments of Figure 18.**
>
>
>
> [r] Weizhi Lu, Mingrui Chen, Kai Guo, and Weiyu Li. Quantization: Is it possible to improve classifcation? In Data Compression Conference, pp. 318–327. IEEE, 2023.
>
> **Furthermore, our findings  should also apply to multi-label classification problems,** since **multi-label** classification is typically achieved by transforming it into binary or multiple classification problems [r].
>
> [r] https://en.wikipedia.org/wiki/Multi-label_classification
>
> **Q5:** The paper provides theoretical analysis that the appropriate threshold $\tau$ exists, but how to set it not depending on the classification accuracy?
>
> **A5:** Thank you. Theoretically, the desired $\tau$ that enhances feature discrimination could be determined by optimizing Equations (8) and (9). However, we encounter difficulties in addressing this problem due to the complex forms of the derivatives of (8) and (9). To address this challenge, we can utilize approximate solution algorithms, such as the simple yet effective bisection method.

---

### Comment · Area_Chair_Kp7M · 2024-11-23
**Please review author response**

Dear reviewer,

Could you review the author response and let them know if you are satisfied with it or if you have any additional questions?

Kind regards,

Your AC

---

### Meta-Review · Area_Chair_Kp7M · 2024-12-23

**Metareview:**

This work investigates how the classification performance is impacted when features are quantized, especially by binary and ternary quantization. It highlights that feature discrimination, instead of quantization error, shall be used to investigate the impact of quantisation for classification. Theoretical analysis is conducted to prove that there exists a quantization threshold that can achieve better feature discrimination after binary or ternary quantization of features. Experimental study is conducted to demonstrate the results of theoretical analysis.

Reviewer aSct is very positive on this work and comments that this work addresses a meaningful problem, proposes novel analysis, has rigorous theoretical proof, excellent explanation, and adequate experiments. Reviewer Ez39 comments that the work is original and interesting and the paper is well presented. Also, Reviewer JS4C comments that theoretical derivations are well supported with numerical experiments. At the same time, the reviewers raise issues related to further clarification on feature discrimination, the applicability of the analysis to more general settings, the effectiveness on multi-class classification, identifying appropriate threshold, the significance of the finding, the design of experiments, the limitations on dataset size, classification tasks, and classifiers investigated.

The authors provide a rebuttal. It is appreciated that further clarifications and more experimental study are provided. However, the rebuttal does not fully address the concerns on the applicability of this work to large deep models (by Reviewer JS4C) and the effectiveness of this analysis for multi-class classification and various classifiers (by Reviewer Ez39). The final ratings are 8, 5, and 5.

AC carefully reviews the submission, the comments, the rebuttals, and the discussion. This work has its merit in theoretically proving that applying binary and ternary quantization of features could even improve feature discrimination, which could lead to better classification performance. Also, this work conducts experiments to show that this result can indeed be observed on synthetic and real datasets. However, this work has the following issues: 1) Although rigorous and inspiring, the analysis is derived under an ideal (or highly simplified and restrictive) setting that only considers Gaussian distribution, same variance, univariate, and separable classes, etc. This setting can hardly be satisfied in practice, especially for complex classification problems; 2) The claim that “this is the first study that exploits feature discrimination to analyze the impact of quantization on classification” needs to be more strongly justified because quantizing features to transform continuous variables into discrete ones is a common step in the field of pattern classification and its effect has been intensively studied in the literature; 3) Although additional experiments are provided in this work to show its applicability to multi-class classification, the study is not systematic or comprehensive and therefore not convincing enough; 4) This work does not give an algorithm to find the optimal quantization threshold in practice; 5) In addition, a less significant issue is that this work regards the KNN as a linear classifier and relies on it to conduct investigation. However, this is flawed because KNN is a nonlinear classifier.

Taking all the factors into account, this work in its current form cannot be recommended for acceptance. It is hoped that the reviews could help to further improve the quality of this work.

**Additional Comments On Reviewer Discussion:**

The reviewers raise issues related to further clarification on feature discrimination, the applicability of the analysis to more general settings, the effectiveness on multi-class classification, identifying appropriate threshold, the significance of the finding, the design of experiments, the limitations on dataset size, classification tasks, and classifiers investigated.

The authors provide a rebuttal. It is appreciated that further clarifications and more experimental study are provided. However, the rebuttal does not fully address the concerns on the applicability of this work to large deep models (by Reviewer JS4C) and the effectiveness of this analysis for multi-class classification and various classifiers (by Reviewer Ez39). The final ratings are 8, 5, and 5.

AC carefully reviews the submission, the comments, the rebuttals, and the discussion. This work has its merit in theoretically proving that applying binary and ternary quantization of features could even improve feature discrimination, which could lead to better classification performance. Also, this work conducts experiments to show that this result can indeed be observed on synthetic and real datasets. However, this work has the following issues: 1) Although rigorous and inspiring, the analysis is derived under an ideal (or highly simplified and restrictive) setting that only considers Gaussian distribution, same variance, univariate, and separable classes, etc. This setting can hardly be satisfied in practice, especially for complex classification problems; 2) The claim that “this is the first study that exploits feature discrimination to analyze the impact of quantization on classification” needs to be more strongly justified because quantizing features to transform continuous variables into discrete ones is a common step in the field of pattern classification and its effect has been intensively studied in the literature; 3) Although additional experiments are provided in this work to show its applicability to multi-class classification, the study is not systematic or comprehensive and therefore not convincing enough; 4) This work does not give an algorithm to find the optimal quantization threshold in practice; 5) In addition, a less significant issue is that this work regards the KNN as a linear classifier and relies on it to conduct investigation. However, this is flawed because KNN is a nonlinear classifier.

Taking all the factors into account, this work in its current form cannot be recommended for acceptance.

---

### Decision · Program_Chairs · 2025-01-22

Reject